# 3D printing of thermosets with diverse rheological and functional applicabilities

Yuxuan Sun[1], Liu Wang [2]✉, Yangyang Ni[1], Huajian Zhang[1], Xiang Cui[3], Jiahao Li[2], Yinbo Zhu[2], Ji Liu [4], Shiwu Zhang[1], Yong Chen [5]✉ & Mujun Li [1]✉

Thermosets such as silicone are ubiquitous. However, existing manufacturing of thermosets involves either a prolonged manufacturing cycle (e.g., reaction injection molding), low geometric complexity (e.g., casting), or limited processable materials (e.g., frontal polymerization). Here, we report an in situ dual heating (ISDH) strategy for the rapid 3D printing of thermosets with complex structures and diverse rheological properties by incorporating direct ink writing (DIW) technique and a heating-accelerated in situ gelation mechanism. Enabled by an integrated Joule heater at the printhead, extruded thermosetting inks can quickly cure in situ, allowing for DIW of various thermosets with viscosities spanning five orders of magnitude, printed height over 100 mm, and high resolution of 50 µm. We further demonstrate DIW of a set of heterogenous thermosets using multiple functional materials and present a hybrid printing of a multilayer soft electronic circuit. Our ISDH strategy paves the way for fast manufacturing of thermosets for various emerging fields.

Thermosets such as silicone-/epoxy-based plastics have been abundantly used in engineering, infrastructure, and daily life in the past century due to their excellent mechanical strength, thermal stability, and chemical resistance[1–3]. Recent advances in emerging fields such as soft robotics and flexible electronics also call for the application of thermosetting components with intricate structures and multi-functionality[4–8]. To date, the manufacturing of thermosets still heavily relies on mold-based methods that allow the material to cure into its hardened forms in autoclaves or ovens, e.g., casting[3,8–10], compression molding[1], and reaction injection molding[3]. However, these conventional methods usually involve long manufacturing cycle time, low geometric complexity, and cost-prohibitive facilities, which are not amenable to fast prototyping of intricate components and have low efficacy in creating heterogeneous and multi-functional devices (Supplementary Table 1).

Recently, additive manufacturing or 3D printing has been emerging as a prominent technology for the rapid manufacturing of thermosets[11–13]. Many strategies have been proposed to realize the 3D printing of thermosets (Supplementary Table 2). Some are utilizing vat polymerization-based 3D printing such as digital light process (DLP)[14–16], stereolithography (SLA)[17], and direct sound printing (DSP)[18]. For example, Kuang et al. reported a single-vat grayscale-DLP method and a two-stage curing ink to obtain functionally graded thermosetting materials[15]. However, DLP and SLA are often limited to light-curable resins and DSP can only print a certain selection of thermosetting inks with specific porosity (by generating sonochemical reactions in the cavitation bubbles). Other approaches for 3D printing thermosets are based on the direct ink writing (DIW) technique[19–22]. Usually, successful DIW requires a yield-stress ink to form self-support. For example, some self-assembled bottlebrush polymers[23,24] can be directly printed. But for many low-viscosity thermosets, DIW is often difficult due to structural collapse. To address this challenge, a widely used solution is to change the low-viscosity thermosetting ink to a yield-stress fluid by adding a rheology modifier such as microsized NaCl particles[25],

[1]Department of Precision Machinery and Precision Instrumentation, University of Science and Technology of China, 230026 Hefei, China. [2]CAS Key Laboratory of Mechanical Behavior and Design of Materials, Department of Modern Mechanics, University of Science and Technology of China, 230026 Hefei, Anhui, China. [3]School of Computer Science and Technology, University of Science and Technology of China, 230026 Hefei, China. [4]Department of Mechanical and Energy Engineering, Southern University of Science and Technology of China, 518055 Shenzhen, China. [5]Epstein Department of Industrial and Systems Engineering, Viterbi School of Engineering, University of Southern California, Los Angeles, CA 90089, USA. ✉e-mail: wangliu05@ustc.edu.cn; yongchen@usc.edu; lmn@ustc.edu.cn

nanosized silica[26–30], and wax[31]. However, the rheological modification inevitably tailors the material properties of the thermoset and requires modification optimization (e.g., particle size, composition, weight fraction) for each thermoset to be successfully printed. In addition, despite the enhanced yield strength, post-curing of thermosetting polymer is usually necessary so that the printed architectures may still have relatively low heights to avoid collapse[20–22,32].

The second strategy for DIW of thermosets is embedded 3D printing that carries out the printing process in a supporting medium[33–36]. This method, although it can print complex structures such as silicone scaffolds[33], often yields low structural integrity (e.g., weak filament-to-filament bonding) and has limitations in printing heterogeneous and multifunctional thermosets[36,37]. Recently, frontal polymerization-assisted DIW of thermosets has also been reported[38–40]. By triggering the ring-opening metathesis polymerization (ROMP) with a catalyst, the extruded ink undergoes in situ gelation process at the nozzle front, allowing for quick solidification of printed thermosets. However, this ROMP-enabled in situ gelation mechanism seems to have been exclusively adopted for manufacturing dicyclopentadiene-based thermosetting composites[40], which does not apply to other types of thermosets. Last but not least, some photo-curable thermosetting inks (even with low viscosity) may also be directly written by utilizing in situ

light-curing during the DIW process. For instance, some liquid crystalline polymers can be printed under UV light[41,42]. Similar to DLP and SLA, however, this 3D printing strategy may only apply to light-curable thermosets. Therefore, a general strategy for the fast 3D printing of diverse thermosets with untailored material properties, high structural integrity and complexity, as well as heterogeneity and multifunctionality has not been reported yet, to the best of our knowledge.

Here, we report an in situ dual heating (ISDH) strategy to achieve rapid 3D printing of thermosets by harnessing the heating-accelerated in situ gelation mechanism (Fig. 1a). By integrating a Joule heater into a printhead, freshly extruded thermosetting inks undergoes quick in situ gelation at an elevated temperature (Fig. 1b), allowing for fast curing of thermosets (shown by the steep increase of the viscosity in Fig. 1c). The ISDH-enabled fast gelation enables DIW of thermosets with a diverse selection of rheological and functional properties (Fig. 1d). We first demonstrate the DIW of a set of complex structures using various thermosetting inks and achieve a maximum height of 120 mm and resolution of 50 μm (Fig. 1e). In the absence of rheology modifiers or auxiliary chemical reactions (except for crosslinking), the printed thermosets preserve identical mechanical properties and chemical characteristics to the cured counterparts by the conventional molding method. We further demonstrate DIW of several heterogeneous

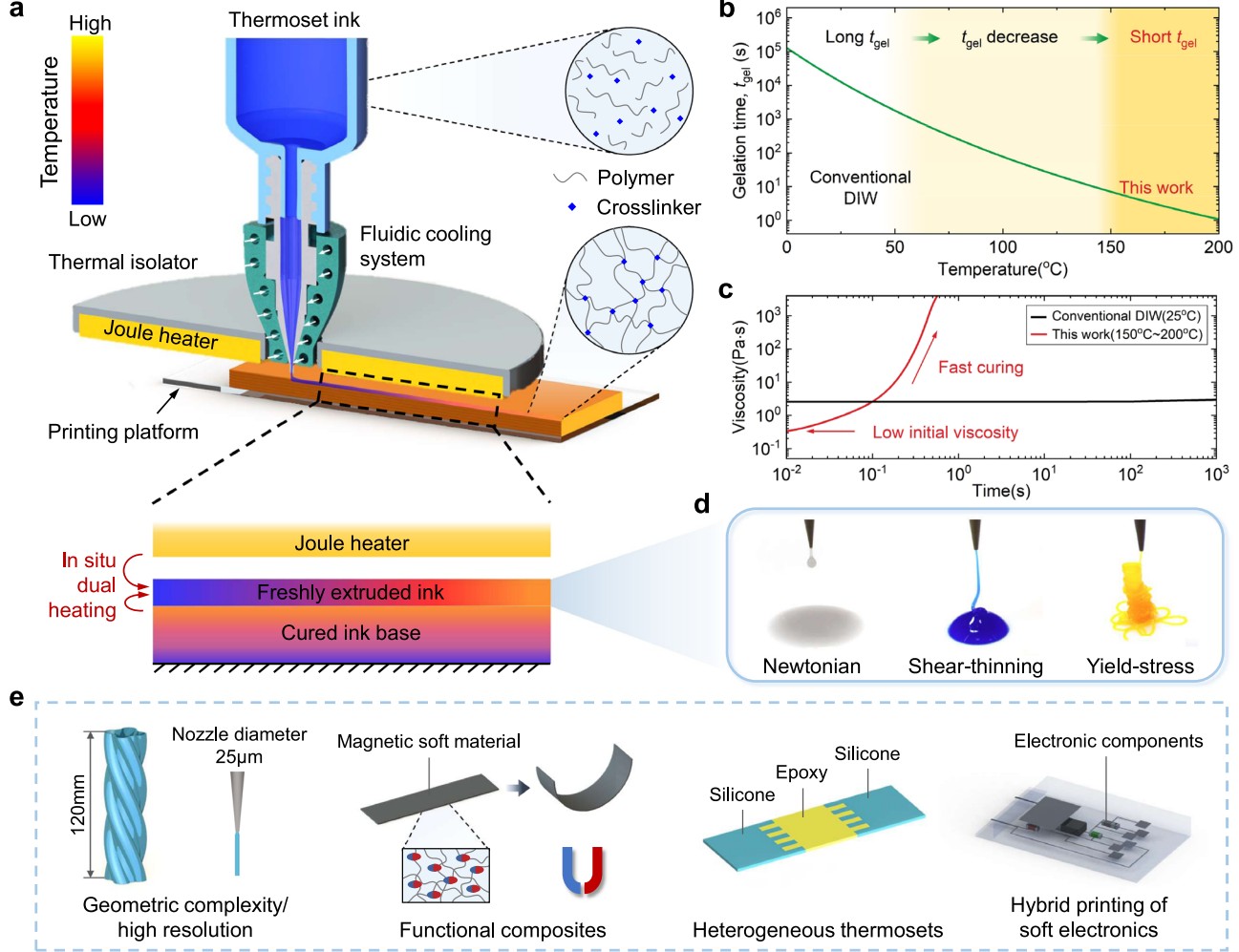

**Fig. 1 | Overview of in situ dual heating-enabled DIW of thermosets. a** Schematic illustration of the DIW platform in which a Joule heater is integrated into the nozzle. The freshly extruded ink undergoes in situ gelation as being heated bilaterally by the Joule heater and the cured ink base, referred to as in situ dual heating (ISDH). **b** The gelation time at different temperatures of a representative thermosetting ink (Sylgard 184). **c** Comparison of viscosity as a function of time between conventional DIW (-25 °C) and our work (150-200 °C). The solid curves are calculated using Eq. (1) by selecting Sylgard 184 and setting the temperature as 25 °C and 190 °C, respectively. **d** Exemplary printable inks with various rheological properties such as Newtonian, shear-thinning, and yield-stress. **e** The ISDH strategy enables DIW of thermosets with high geometric complexity and resolution. Functional composites, heterogeneous thermosets, and soft electronics can also be readily printed.

thermosets using multiple materials (e.g., magnetic composites with different particle concentrations) and present a hybrid printing of a multilayer soft electronic circuit.

## Results

### ISDH-enabled fast gelation

The underlying principle of ISDH-enabled DIW technology is the heating-accelerated gelation of the thermosetting ink. The schematic illustration of ISDH printing platform is shown in Fig. 1a and Supplementary Fig. 1 in which a Joule heater is integrated into the printhead to create a high-temperature field. The introduction of the Joule heater has a twofold effect. First, it continuously heats the upper side of the freshly extruded ink; Second, it keeps the top surface of the already cured ink base at a high temperature. Therefore, when the ink is extruded and deposited onto the cured ink base, its upper and lower sides are simultaneously heated in situ, which is referred to as in situ dual heating. The thermosetting ink remains at a low temperature in the nozzle by employing a heat isolator and a fluidic cooling system. After being extruded and deposited, it undergoes a fast in situ gelation process due to the drastic increase in temperature, and thereafter serves as the base for supporting the next layer. As shown in Fig. 1b, while previous works conducted at room temperature involve a long gelation time $t_{gel}$, in the ISDH-enabled DIW, the quickly raised temperature decreases $t_{gel}$ to approximately several seconds, allowing for fast solidification of printed filaments. Such a quick in situ gelation process can be readily reflected in the steep increase in the viscosity (denoted as $\eta$, Fig. 1c), which can be characterized by the dual-Arrehenius rheology model[43]:

$$\ln \eta(t) = \ln \eta_\infty + E_{a,\eta}/RT + \int_0^t k_\infty e^{-E_{a,k}/RT} dt \qquad (1)$$

where $\eta(t)$ is the time-dependent viscosity under a constant temperature $T$, $\eta_\infty$ is the calculated viscosity at $T = \infty$, $E_{a,\eta}$ is the activation energy for viscosity, $R$ is the ideal gas constant, $k_\infty$ is calculated apparent kinetic factor at $T = \infty$, and $E_{a,k}$ is the activation energy for gelation. A detailed modeling of gelation is provided in Supplementary Text. In brief, on the one hand, the instantaneous viscosity drops at higher temperatures (characterized by the first two terms $\ln \eta_\infty + E_{a,\eta}/RT$). On the other hand, the viscosity escalates due to heating-accelerated crosslinking (characterized by the third term $\int_0^t k_\infty e^{-E_{a,k}/RT} dt$). Take a typical thermosetting material, silicone Sylgard 184, as an example. As a two-part silicone, the mixing ratio of Part A (base monomer) and Part B (crosslinker) (compositions are shown in Supplementary Fig. 2) will change the viscosity of Sylgard 184. By default, Sylgard 184 in this work refers to that with Part A: Part B = 10:1 mass ratio unless other mixing ratios are specified. The time-dependent viscosity at $T = 25\,°C$ and $190\,°C$ are plotted in Fig. 1c, respectively. Sylgard 184 is usually considered a Newtonian fluid at $T = 25\,°C$ and it takes approximately 30 h to reach the gel point (crosslinking extend $p = 66\%$ for silicone). When the temperature rises to $190\,°C$, the gelation time decreases to 1.05 s (viscosity increases more than three orders of magnitude within 1 s), transferring the flowable liquid to a solid. The fast gelation can be further reflected by the significantly reduced energy barrier for crosslinking reactions at $190\,°C$ according to density functional theory (DFT) simulation (Supplementary Fig. 3).

To elucidate the ISDH mechanism, we label the freshly extruded ink/filament and the top surface of the cured ink base as $(i+1)^{th}$ and $i^{th}$ layer, respectively (Fig. 2a). The temperature of Joule heater, $(i+1)^{th}$ layer, and $i^{th}$ layer is denoted as $T_{heater}$, $T_{i+1}$, and $T_i$, respectively. When the ink is extruded and deposited on the cured ink base, its upper side is heated by the Joule heater via convection and radiation, while its lower side is heated by the $i^{th}$ layer via conduction. The temperature of the $(i+1)^{th}$ and $i^{th}$ layers are measured using an infrared camera during

printing (Supplementary Fig. 4). By setting $T_{heater} = 260\,°C$, the $i^{th}$ layer can maintain an almost constant temperature $T_i = 190\,°C$ regardless of the printing height (e.g., $T_i \equiv 190\,°C$ for $i = 10, 12, 15$ when $T_{heater} = 260\,°C$ in Fig. 2b). Therefore, for any freshly extruded ink, its temperature can quickly increase to $T_{i+1} = 196\,°C$ at $t = 2$ s after being deposited on the $i^{th}$ layer. To further investigate the heat transfer inside the freshly extruded ink, we conduct finite element analysis (FEA) using the commercial software COMSOL Multiphysics. The temperature change and heat flux through the upper and lower side of the $(i+1)^{th}$ layer are presented in Fig. 2c. Initially, the freshly extruded ink (i.e., $(i+1)^{th}$ layer) has a temperature of $25\,°C$. When deposited on the cured ink base, its lower side is quickly heated up (see the temperature contour plot at $t = 0.5$ s) by the $i^{th}$ layer. During this period, the heat flux from the lower side is one order of magnitude larger than that through upper side from the Joule heater (Fig. 2d). After being heated to $190\,°C$, heat flux through the lower side attenuates, while the freshly extruded ink is continuously heated by the Joule heater. Note that the $(i+1)^{th}$ layer has a relatively uniform temperature distribution at $t = 2$ s in which the upper side is $196\,°C$ and the lower side is $193\,°C$ (see the temperature contour plot at $t = 2$ s in Fig. 2c), giving rise to consistent gelation inside the freshly extruded ink. Thereafter, the cured $(i+1)^{th}$ layer will serve as the base for supporting next layer of freshly extruded ink.

The continuous heating from Joule heater keeps the top surface of the cured ink base at a constant temperature. For example, $T_i \equiv 190\,°C$ when $T_{heater} = 260\,°C$. When $T_{heater}$ is changed, $T_i$ will be altered accordingly (Fig. 2e) and if the Joule heater is removed, $T_i$ will also rapidly decrease (Fig. 2f). Both $T_{heater}$ and $T_i$ affect the aspect ratio of the single filament's cross-section (denoted as h/d in Fig. 2a). We measured the aspect ratio h/d of printed filament at different $T_{heater}$ and $T_i$ values and results are presented in Fig. 2g. It shows that h/d increases when $T_{heater}$ and $T_i$ increases due to the faster gelation. When $T_{heater}$ or $T_i$ is too low, on the one hand, the extrude ink spreads and cannot form a filament of h/d > 0.05, which is undesired for DIW-based 3D printing. On the other hand, when $T_{heater}$ or $T_i$ is too high (e.g., $220\,°C$), air may be trapped between filaments because the gelation time is too short for the bubble to escape (Fig. 2g and Supplementary Fig. 5). Computational fluid dynamics (CFD) simulation indicates that it takes a long time (17 s) for a bubble at the bottom of the filament to escape at a temperature of $50\,°C$, while the bubble can quickly escape when the temperature is $190\,°C$ before the thermosetting ink fully cures (Supplementary Fig. 5). However, if the temperature gets even higher (e.g., $220\,°C$ in Fig. 2h), the ink cures too fast so that the bubble is trapped inside. To validate the CFD results, we choose two nozzles with diameter of 300 um and 1 mm. It is found that no bubbles are observed at $190\,°C$ while bubbles are trapped at $220\,°C$ for both nozzles. Therefore, a good choice for printing filaments with high aspect ratio (h/d > 0.2, green pixels in Fig. 2g) and reduced bubbles is to use $T_{heater} = 240 \sim 260\,°C$ (corresponding $T_i = 170 \sim 190\,°C$). Also, to achieve consistent structural integrity, we suggest continuous heating from Joule heating. Otherwise, reheating the cured ink base is necessary if the heating is interrupted.

### DIW of low-viscosity thermosets

Enabled by in situ fast gelation, we can directly print low-viscosity thermosetting ink into complex 3D structures. Here, we select Sylgard 184 as a representative example. As shown in Fig. 3a, b, a 120-mm high tube is first printed with $T_{heater} = 260\,°C$ (Supplementary Movie 1). The scanning electron microscope (SEM) image shows a solid structure without any voids or cavities. A straight and uniform thin wall is formed without any collapse and distortion, except that the bottom of the tube is a little thicker due to attenuated gelation when the freshly extruded ink contacts the cool printing platform ($25\,°C$) at the beginning of printing (see inserts in Fig. 3b). The degree of cure of ISDH-printed Sylgard 184 is measured as 99.4% according to differential scanning

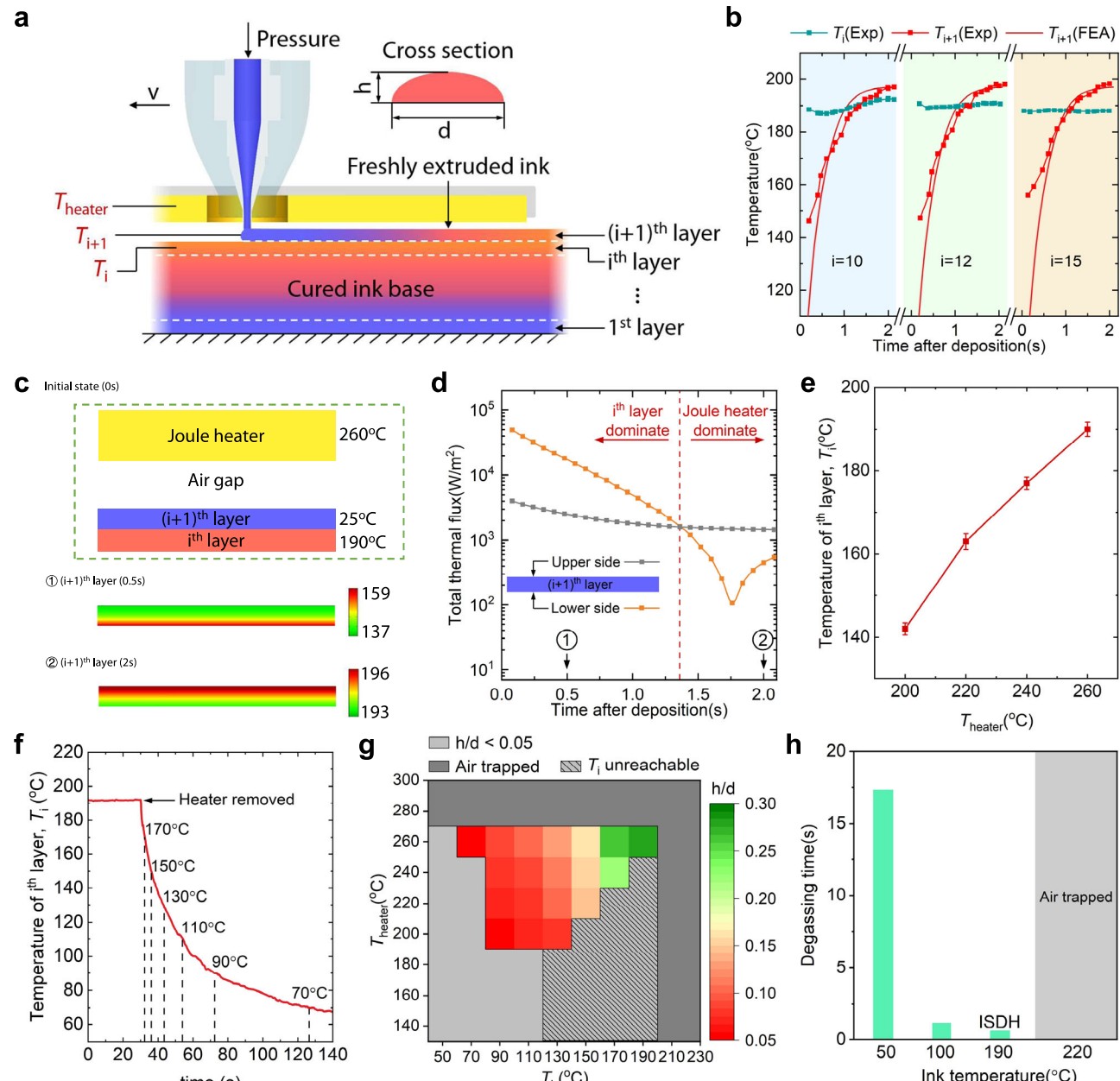

**Fig. 2 | The mechanism of in situ dual heating. a** Schematic illustration of ISDH. The freshly extruded ink is labeled as $(i+1)^{th}$ layer and the top layer of the cured ink base is labeled as $i^{th}$ layer. The temperature of the Joule heater, $(i+1)^{th}$ layer, and $i^{th}$ layer is denoted as $T_{heater}$, $T_{i+1}$, and $T_i$, respectively. **b** Simulated degassing time at 50, 100, 190, and 220 °C. **c** Experimental and simulated temperature of freshly extruded ink when $i=2, 5, 10$. **d** Finite element analysis of the thermal flux on the upper and lower side of the freshly extruded ink. **e** $T_i$ as a function of $T_{heater}$ during continuous heating. **f** Temperature of $i^{th}$ layer as a function of time after the Joule heater is removed. **g** The aspect ratio of the filament's cross-section h/b is plotted as a function of $T_{heater}$ and $T_i$ using Sylgard 184. The unreachable domain refers to $T_i$ greater than the corresponding value in **e** under the same $T_{heater}$. **h** Degassing time under different temperatures.

calorimetry (DSC, Supplementary Fig. 7). An inclined angle of 38° with a wall height of 4 mm can be printed without an additional supporting medium (Supplementary Fig. 8). In the absence of rheology modifiers or auxiliary reactions (except for crosslinking), ISDH-enabled 3D printing is scalable when the printing nozzle varies from 25 μm to 1 mm. For example, a wrinkled ring with a diameter of 90 mm can be successfully printed using a 1-mm nozzle (Fig. 3c) and a filament with a width of 50 μm can be also realized using a nozzle with a 25-μm diameter (Fig. 3d). It is worth noting the ISDH-printed samples have almost the same mechanical properties as the counterparts manufactured by convention molding method. To validate this, dogbone specimens with printing direction parallel and perpendicular to the loading direction are printed (Supplementary Fig. 9) and their

stress–strain curves under uniaxial loading are provided in Fig. 3e. Figure 3e clearly shows that the printed samples have the same mechanical properties in both directions, manifesting an isotropic printing performance. Compared with the molded counterpart, ISDH-printed samples show almost the same Young's modulus and ultimate strength. The reasons for such molding-comparable mechanical properties are twofold. First, ISDH-enabled 3D printing only accelerates the gelation process while the chemical characteristics of the cured samples remain the same as that of the molded counterparts according to infrared and Raman spectroscopy (Fig. 3g, h). Second, the high temperature creates a strong covalent bond between filaments without bubbles (Supplementary Fig. 6). The Si-OH groups of silicone are thermally active, and both chain scission and crosslinking take

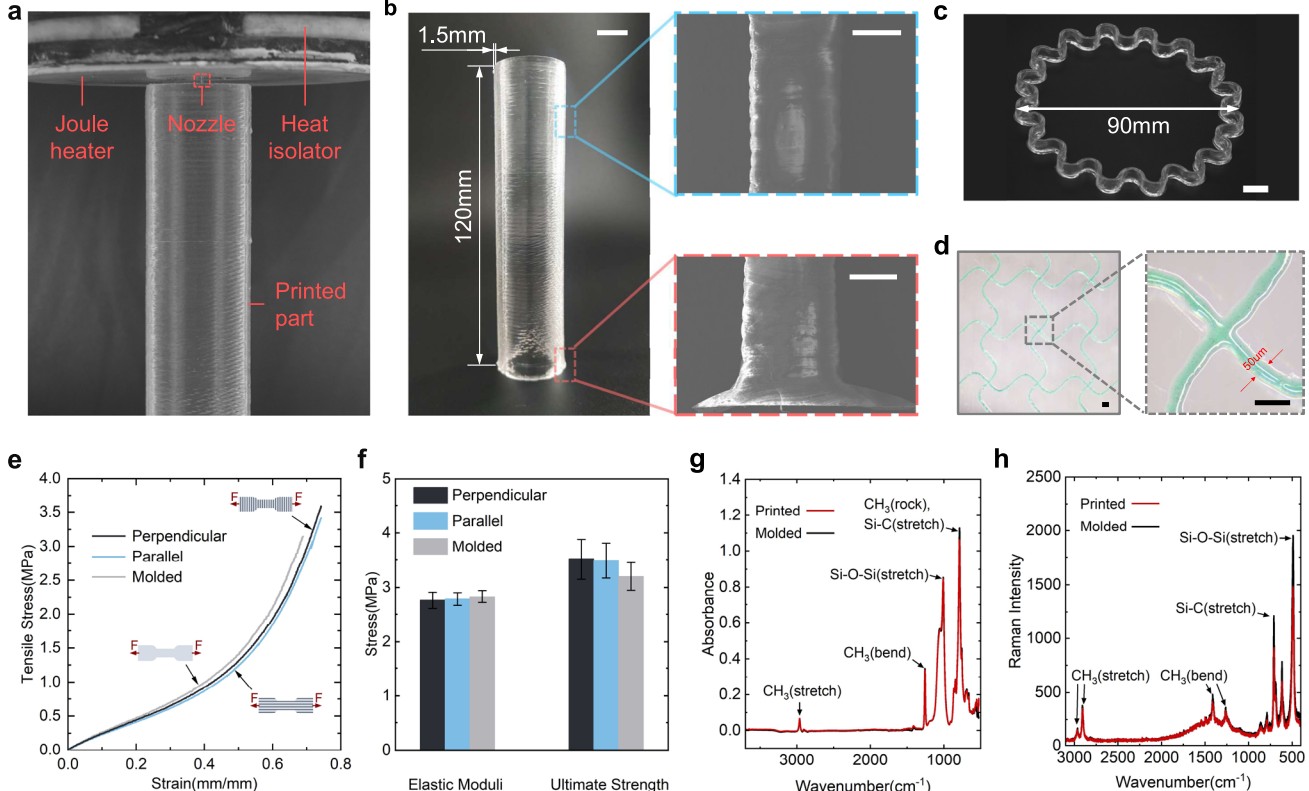

**Fig. 3 | Characterization of ISDH-enabled DIW of low-viscosity Sylgard 184 with base-to-curing agent ratio of 10:1. a** Photograph of printing a cylinder tube. **b** Optical and scanning electron microscope (SEM) images of a printed cylinder tube with a height of 120 mm and thickness of 1.5 mm. (Left scale bar = 12 mm, right scale bar = 0.8 mm). **c** Optical image of a printed wrinkled ring with a diameter of 90 mm using a nozzle with 1-mm diameter (scale bar = 2 mm). **d** Serpentine network printed with a 25 μm nozzle (scale bar = 100 μm). **e–g** Comparison of the **e** tensile test; **f** elastic modulus and ultimate strength; **g** infrared and **h** Raman spectroscopy between ISDH-printed and molded samples.

place at a temperature of 190 °C between crosslinked silicone and uncured silicone. Thus, covalent bonds can still be formed at the filament interface to create strong bonding, yielding a high structural integrity of ISDH-printed samples[44]. In addition, we also show that ISDH-printed samples have a lower surface roughness than that printed by adding silica rheology modifiers (Supplementary Fig. 10). This result is expected as our method does not employ any other additives. The ISDH strategy can also directly print Sylgard 184 with different ratios of Part A to Part B (e.g., 2:1, 5:1, 20:1) and yields molding-comparable quality. By comparing the infrared and Raman spectroscopy in Supplementary Fig. 11, we show that all printed samples have the same chemical characteristics as that of the molded counterpart. Particularly, for the 2:1 Sylgard 184, a peak at 2164.6 cm⁻¹ is observed for both ISDH-printed and molded samples, which corresponds to the Si-H stretch, indicating the existence of excessive and uncrosslinked monomers. Notably, Fig. 3e–h and Supplementary Fig. 11 also suggest that no oxidization occurs during the ISDH-enabled 3D printing at 190 °C. To further validate this point, we intentionally made an oxidized sample of Sylgard 184 (cured temperature is 250 °C) and found that it has different mechanical properties and infrared spectroscopy results (e.g., a peak at 1650 cm⁻¹ shows a C = O group) (Supplementary Fig. 12).

### DIW of thermosets with diverse rheological properties

The ISDH-enabled DIW technology can print thermosetting inks with a broad range of rheological properties (Fig. 4a, b) into complex 3D shapes. Demonstrations are shown in Fig. 4c–k where selected inks include Sylgard 184, Sylgard 186, DragonSkin 30, DragonSkin 10, DragonSkin 10 + SiO₂ (20 wt% nanoparticles rheology modifier), Ecoflex 00-30, Ecoflex 00-10, and epoxy. Rheological measurements show

that different thermosetting inks possess dynamic viscosity changes across five orders of magnitude (Fig. 4a). Yield stress is only observed in DragonSkin 30, DragonSkin 10, and DragonSkin 10 + SiO₂. For other inks, the yield stress is not observed and thus they cannot be directly printed using conventional DIW technology (Fig. 4b). Note that only representative inks are shown in Fig. 4a, b while rheological measurements of other inks are given in Supplementary Fig. 13. For different materials, their thermo-rheological properties are first measured by a rheometer and DSC. Then, the temperature for printing different materials is roughly determined by keeping the flowable window at approximately 1 s according to Eq. (1). For example, epoxy has an almost constant viscosity of 230 Pa·s and it quickly cures within 1 s upon heating ($T_{heater}$ = 240 °C, $T_i$ = 170 °C) according to DSC measurement (Supplementary Fig. 14).

### DIW of heterogeneous and functional thermosets

In addition to the diverse rheological properties, the ISDH-enabled DIW technology can also manufacture heterogeneous and functional thermosets. First, we show that a heterogeneous design of USTC logo can be printed with different inks (DragonSkin 10, Sylgard 184, and DragonSkin 10 + SiO₂) in Fig. 5a. Note that the ink is changed after each part is printed. The USTC logo can be further stretched without interfacial debonding. Using Ecoflex 00-30 and epoxy, we can also print a heterogeneous thermoset with interlocked interfaces (Fig. 5b). This heterogeneous thermoset exhibit distinct material properties: the Ecoflex part is highly deformable while the epoxy part is very rigid. To demonstrate the ability of design-oriented functional thermosets, we print a set of magnetically responsive architectures using magnetic composites (NdFeB microparticles embedded in Sylgard 184 matrix) with variable magnetic properties. By tuning the volume fraction of embedded NdFeB

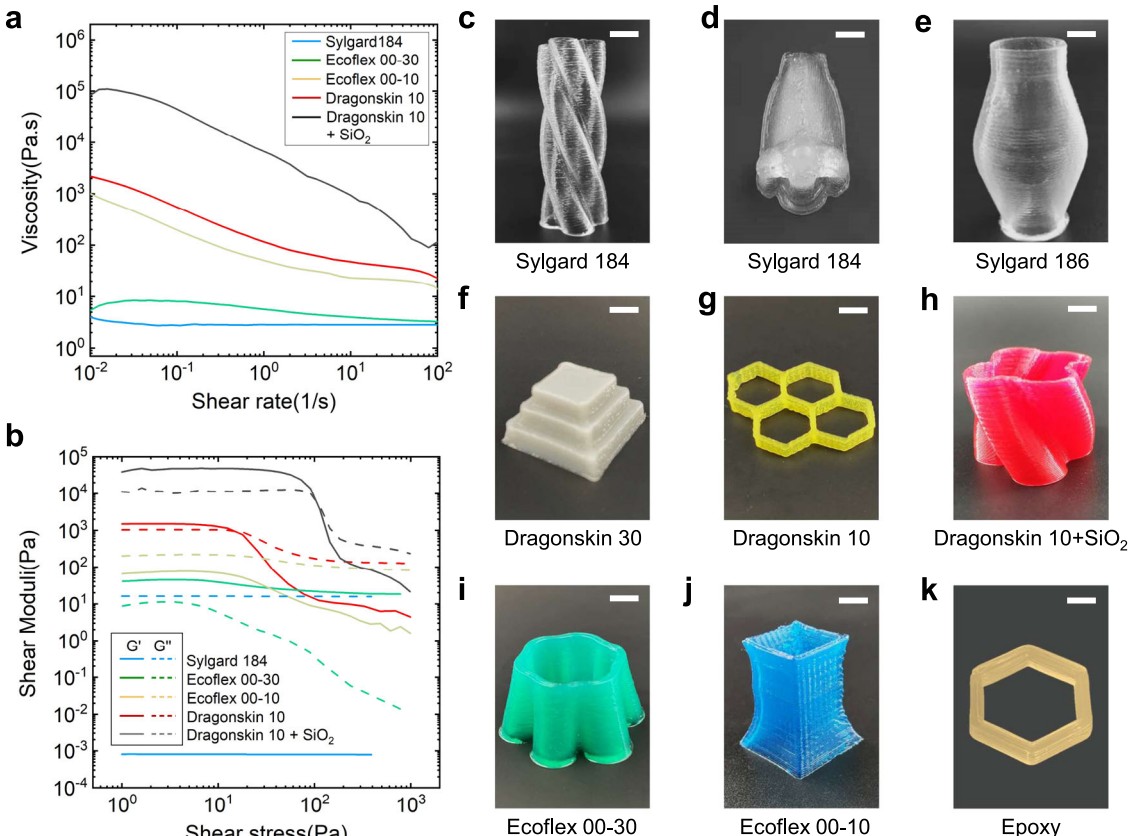

**Fig. 4 | Demonstration of ISDH-enabled DIW of diverse thermosets.**
**a**, **b** Viscosity and shear moduli of some typical thermosetting inks with variable rheological properties. **c**–**k** ISDH-printed parts using **c**, **d** Sylgard 184; **e** Sylgard 186; **f** DragonSkin 30; **g** DragonSkin 10, **h** DragonSkin 10 + 20%wt SiO₂; **i** Ecoflex 00-30; **j** Ecoflex 00–10; **k** Epoxy. Scale bar = 15 mm for **c**, **d**, scale bar = 8 mm for **e**–**g** and scale bar = 5 mm for **h**, **k**.

(i.e., 0, 5, 10, 15, 20 vol%), magnetic composites have different rheological properties (Supplementary Fig. 15). The ISDH-enabled DIW offers a free combination of different magnetic composites and three representative magnetic architectures are displayed in Fig. 5c−e. First, a five-segment magnetic strip is printed in which each segment has a specific NdFeB vol% (e.g., 0, 5, 10, 15, 20%) (Fig. 5c). The magnetization directions are denoted by the black arrows. When actuated by a uniform magnetic field ($B = 50$ mT), it shows unique deformation patterns due to the programmed magnetic and mechanical properties in each segment (Supplementary Movie 2). Second, a magnetic flower with three petals (vol% = 5, 15, 20%) is shown in Fig. 5b. Three petals show different deflecting angles when the magnetic field is applied. Third, the ISDH-enabled DIW can also print functional thermosets with intricate 3D structures. For example, magnetic stents that have promising applications in cardiovascular diseases have been reported in some recent works[45,46]. Due to the hollow thin wall and highly tilted connecting ribbons, conventional molding and DIW approaches have difficulty or low efficacy in their manufacturing. Distinctly, we can directly print the magnetic stent using our ISDH-enabled DIW technology as shown in Fig. 5e. The printing process is given in Supplementary Movie 3. When a magnetic field is applied, it can have lateral shrinkage that agrees well with FEA results.

### Demonstration of hybrid printing of soft electronics

The in situ fast gelation not only provides self-support for constructing high complexity structures but also the ability to incorporate prefabricated parts or components to create functional devices. The heating temperature of ISDH is below the welding temperature of most electronic components, making it suitable to fabricate soft electronics. By assembling pick-and-place electronic components with 3D printing

(referred to as hybrid printing of soft electronics[47]), we present two demonstrations of hybrid printing of soft electronics. First, we print a conductive coil with pick-and-place LED light on a soft Ecoflex substrate (Fig. 6a, b). The LED light can be tuned on by remotely applying an alternating magnetic field (Fig. 6c), even when the soft Ecoflex substrate is deformed (Fig. 6d). Second, we present a multilayer soft touch sensor (Fig. 6e). Detailed fabrication process is shown in Supplementary Fig. 16. Briefly, after printing the first layer of Ecoflex, stretchable conductive inks are printed to form electronic circuits and pads. The second layer of Ecoflex is then printed with reserved holes for embedding pick-and-place electronics. Thereafter, the third layer of Ecoflex is printed as an insulating medium. Another layer of conductive ink circuit and Ecoflex encapsulating layer is subsequently printed. Note that conductive inks are printed to the reserved holes to form interlayer conduction. Figure 6f shows the printed soft touch sensor, where a power indicator (red LED) is used to identify the power supply and a touchpad is adopted to sense the capacity change that is reflected by the target LED (green). Connecting to the power source will light the power indicator (Fig. 6g); when touching the touchpad with fingers, the capacitance increases, thereby turning on the target LED (Fig. 6g and Supplementary Movie 4). The ISDH-enable hybrid printing allows for high structural integrity with strong interfacial bonding. After being stretched, bent, and twist (Fig. 6h), the soft touch sensor can still work.

### Discussion

By integrating a Joule heater into the printhead, we develop a 3D printing platform that is capable of DIW of thermosets with diverse rheological and functional applicabilities. The Joule heater provides a high-temperature environment that keeps the top surface of the cured

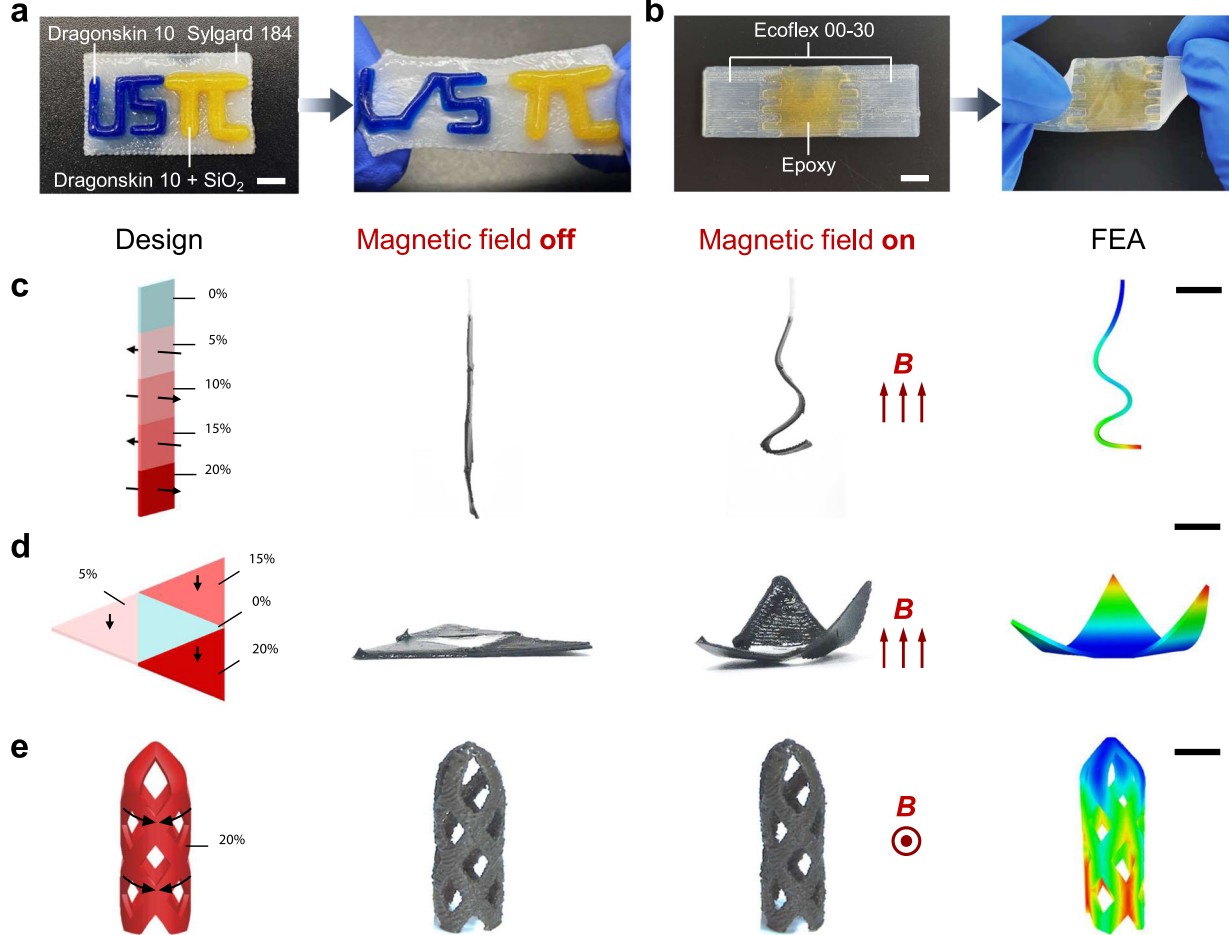

**Fig. 5 | DIW of heterogeneous and functional thermosets. a** Heterogeneous thermosets using Sylgard 184, DragonSkin 10, and DragonSkin 10 + SiO₂ (scale bar = 5 mm). **b** Heterogeneous thermosets using epoxy and Ecoflex 00-30 (scale bar = 5 mm). **c** A magnetic strip with five segments in which NdFeB vol% varies from 0 to 20%. The magnetization directions are denoted by black arrows. **d** A magnetic flower with three petals in which NdFeB vol% are 5, 15, and 20%. The magnetizations are pointing down. **e** A magnetic stent printed with 20 vol% magnetic composites. The magnetizations are in the hoop direction as highlighted by black arrows. Simulated deformed shapes agree well with experiments when the magnetic field of 50 mT is applied (scale bar = 8 mm).

ink base at a high temperature. When the thermosetting ink is freshly extruded and deposited, both its upper and lower sides are heated up, leading to ISDH-enabled fast crosslinking reaction. As widely known that heating facilitates chemical reactions, we show that heating reduces the energy barrier for crosslinking according to DFT simulations, giving rise to rapid DIW of various thermosetting materials with complex geometries, heterogeneous structures, and multifunctionality. Without rheology modifiers and additional chemical reactions (excerpt for crosslinking), ISDH-printed samples have the same mechanical properties and infrared/Raman spectroscopy as that of molded counterpart. Offering a new strategy for the rapid manufacturing of thermoset, our ISDH-enabled DIW technology may also be combined with other manufacturing approaches for more possible applications in the future. Inspiring works are from Qi and colleagues who have demonstrated 3D printing of complicated structures by combining DIW and DLP techniques[48,49]. In this regard, the printing temperature, as the most critical parameter in our work, should be carefully selected for each part to be successfully printed. Possible side effects such as materials decomposition, oxidization reaction, and bubble trapping should be avoided.

## Methods
### Ink preparation
**Pure thermosetting ink.** Sylgard 184 and Sylgard 186 are purchased from Dow Corning, USA. DragonSkin 30, DragonSkin 10, Ecoflex 00-30, and Ecoflex 00-10 are purchased from Smooth-on, USA. The silicone ink was prepared by directly blending Part A and B at a specific mass ratio in a 50 ml beaker. For Sylgard 184, Part A: B = 10:1 is used unless other mixing ratios are specified. For Slygard 186, Part A:B = 10:1. For DragonSkin 30, DragonSkin 10, Ecoflex 00-30, and Ecoflex 00-10, Part A:B = 1:1. For DragonSkin 10 + 20 wt% SiO₂ and blended using a planetary mixer (ARV-310, Thinky, Japan) at 2000 rpm for 5 min. The epoxy ink is purchased from Sinoepc China, and mixing E39D and E20 with 1:1 weight ratio by manual stirring under heating with a heat gun, then added 14.2 phmr 2-ethyl-4-methylimidazole (Aladdin, China) as curing agent. For all these materials, colorants are added for identification.

**Magnetic composite ink.** The magnetic composite ink is prepared by adding NdFeB microparticles with an average size of 5 μm (MQFP-B+, Magnequench, Canada) into Sylgard 184 with different volume ratios and blended using a planetary mixer (ARV-310, Thinky, Japan) at 2000 rpm for 5 min.

**Conductive Ink.** The conductive ink is prepared by adding Ag flakes (100*2 μm, Shanghai Xinzuan Alloy Material Co., Ltd., China) and fluroelastomer (FPM) (product number: G801, DAI-EL, Japan) to DMC (product number: 517127, anhydrous, ≥99%, Sigma–Aldrich, USA) with a weight ratio FPM: DMC: Ag = 1:7:3 and then mixed using a planetary mixer (ARV-310, Thinky, Japan) at 2000 rpm for 30 min.

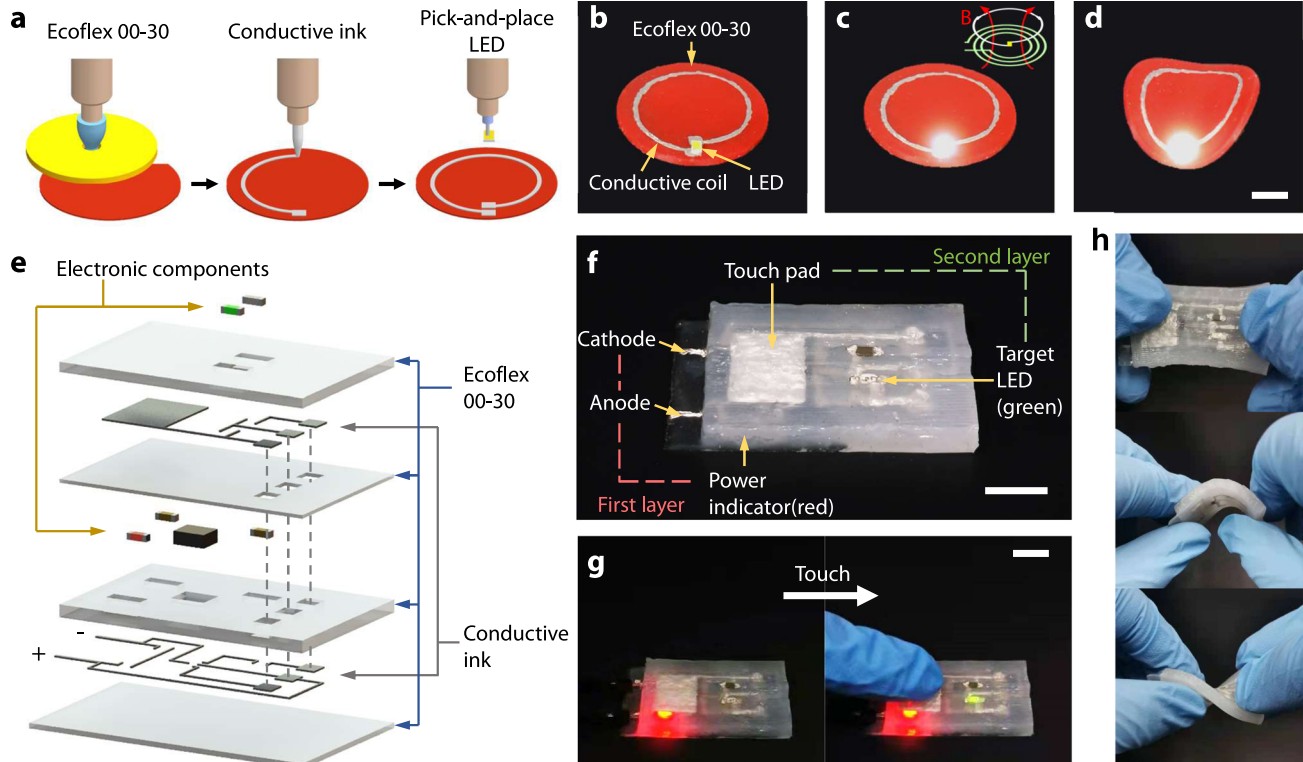

**Fig. 6 | Demonstration of hybrid 3D printing of soft electronics. a** Schematic illustration of printing a conductive coil with pick-and-place LED on Ecoflex 00-30 substrate. **b** Image of the conductive coil with pick-and-place LED on Ecoflex 00-30 substrate. **c** The LED light can be turned on by an alternating magnetic field. **d** The LED light keeps on when the Ecoflex substrate is bent. **e** Schematic design of a multi-layered soft touch sensor. **f** Image of the soft touch sensor consisting of a capacitive touchpad, LED light, and power indicator (scale bar=8 mm). **g** The LED light turns on when a finger touches the soft touch sensor (scale bar = 10 mm). **h** The soft touch sensor reserves high structural integrity under stretching, bending, and twisting deformation.

## ISDH 3D printing procedure

**ISDH printing of pure thermosetting inks.** The ISDH 3D printer is adapted from a commercial FDM printer (Ultimaker 2+ Extended, Ultimaker, Netherlands). A ceramic ring-shaped Joule heater with an embedded K-type thermocouple is mounted to the printhead. A thermal isolator is covered on the heater to reduce heat dissipation. A liquid chiller is customized and fabricated by selective laser sintering (SLS) technology to fit the nozzle shape. Before printing, all the as-prepared inks were transferred into 30 cc syringe barrels (PSY-30F, MUSASHI engineering, Japan) and centrifuged at 2200 rpm for 5 time to remove bubbles. Then, the syringe barrels were connected to a digital pneumatic regulator (SuperΣ CMIII, MUSASHI engineering, Japan) and mounted to the printhead. Nozzles with a 25 μm (XYMtech, China), 300 μm and 1 mm inner diameter (MUSASHI engineering, Japan) were used for demonstrations. After turning on the Joule heater and chiller for 5 minutes for thermal balance, distances between the Joule heater, nozzle tip, and base plate are carefully calibrated to $H_{heater} = 1$ mm and $H_{nozzle} = 0.08$ mm (Supplementary Fig. 1b). Printing paths were generated by self-designed STL models (SolidWorks, Dassault Systèmes, France) and converted into G-code by the commercial software Ultimaker Cura (Ultimaker, Netherland). Precleaned glass plates are used as platform for printing.

**ISDH printing of magnetic thermosets.** Magnetic composites with different NdFeB volume fraction were printed subsequently by changing inks after each part is printed. After printing, the magnetic thermosets were first folded into a predesigned shape and then magnetized by an impulse magnetic field (2.8 T) by an impulse magnetizer (M20-2020, HLT Co. LTD, China).

**Hybrid printing of soft electronics.** The soft electronics was designed using the commercial software Altium Designer (Altium Limited, Australia) and then transferred into printing paths. Pick-and-place electronic components are directly pick and placed at designated areas.

## Characterization

**Rheological properties.** The rheological properties of the inks were measured by a rotational rheometer (Physical MCR 301, Anton Paar, Austria). Before each test, the inks were kept isothermally at 25 °C for 5 min. During the viscometry measurements, the shear rate was swept from 0.01 s$^{-1}$ to 100 s$^{-1}$. During the oscillatory measurements, shear stress was swept from 1 Pa to 1000 Pa at 1 Hz.

**DSC measurement.** The crosslinking properties of the inks were measured by differential scanning calorimetry. The inks were first kept isothermally at 25 °C for 5 min and then heated to 200 °C at different rates under nitrogen flow (10, 20, 25, and 30 °C min$^{-1}$).

**Mechanical properties.** The specimens for mechanical tests were printed or cast into a dogbone specimen with a cross-section of 2 mm × 10 mm. Tensile properties were measured on a universal testing machine with a constant crosshead speed of 10 mm s$^{-1}$ (AGS-X, SHIMADZU, Japan).

**Infrared/Raman spectroscopy.** The image of the printed filament was measured using a scanning electron microscope (Hitachi, SU8220, Japan). The surface quality of the printed filament was characterized by an atomic force microscope (Dimension Icon, Bruker, Germany). The cross-section of the printed filaments was measured using an optical microscope (DSX510, Olympus, Japan). All optical images and videos in this study were recorded using a CMOS camera (acA2440-20 gm/gc, Basler, Germany). The infrared image and video for temperature

measurement were recorded using an infrared camera (ETS320, FLIR, USA). The temperature was recorded using FLIR Tools+ software.

## Numerical simulations

**Heat transfer.** Heat transfer of the freshly extruded ink during the ISDH printing process was conducted using the commercial software COMSOL Multiphysics. The thermodynamics coefficients of Sylgard 184 and air are adopted from reference[42,43] and are listed in Supplementary Table 3.

**CFD simulations.** CFD simulation for degassing was conducted using the commercial software Fluent. A user-defined function (UDF) was developed to describe the viscosity of Sylgard 184 under different conditions according to Eq. (1) with coefficients listed in Supplementary Table 4.

**FEA of magnetic composites.** The deformation of the NdFeB magnetic composites under a uniform magnetic field was simulated in the commercial package ABAQUS. To account for the interaction between magnetic composite with embedded NdFeB microparticles with the uniform magnetic field, we adopt a user's subroutine developed by Zhao et al.[50]. In brief, the magnetic torque density $\tau = M \times B$ can be implemented by computing the magnetic Cauchy stress $\sigma^{magnetic} = -B \otimes FM$ where $F$ is the deformation gradient and operator $\otimes$ represents a dyadic product that takes two vectors to yield a second-order tensor. The Young's modulus and magnetization of the magnetic composite with different NdFeB vol% are calculated according to ref. [51]. Details are provided in the Supplementary Text.

**DFT simulations.** DFT models of molecules (base monomer and crosslinker) were built using GAUSSVIEW6[52]. All calculations were done with GAUSSIAN09W program. The DFT method at the B3LYP/6-31G* level of theory was applied in transition states search, intrinsic reaction coordinate (IRC) calculation and geometry optimism. B3LYP/6-311G** level of theory was applied in single-point energy calculation. The transition states were obtained using QST3 method and the stationary points were confirmed by calculation of the energy second derivatives with respect to atomic coordinates. The thermodynamics were calculated using Shermo[53].

## Data availability

The authors declare that the main data supporting the findings of this study are available within the article and its Supplementary Information files. Extra data are available from the corresponding author upon request.

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

## Acknowledgements

This work was supported by the National Natural Science Foundation of China (Grant Nos. 51475442, 12272369), the National Key Research and Development Program of China (Grant No. 2020YFA0710100), the Natural Science Foundation of Anhui Province (Grant No. 2108085ME170), and the Joint Funds from Hefei National Synchrotron Radiation Laboratory (Grant No. KY2090000068) and partially carried out at the USTC Center for Micro and Nanoscale Research and Fabrication and Engineering Practice Center, University of Science and Technology of China.

## Author contributions

Conceptualization: Y.S., L.W., M.L. Methodology: Y.S., M.L., Y.C. Software: H.Z., X.C, J. Li, Y.Z. Validation: Y.S., Y.N., H.Z., J. Li. Investigation: Y.S., Y.N., H.Z., J. Li, Y.Z. Writing—original draft: Y.S., L.W. Writing—review and editing: L.W., S.Z., Y.C., M.L., J. Liu. Visualization: Y.S., L.W., Y.N., M.L. Supervision: L.W., M.L., Y.C.

## Competing interests

The authors declare no competing interests.
