## [Peer Review File · Nature Communications]

3D printing of thermosets with diverse rheological and functional applicabilitiesReviewers' Comments:

Reviewer #1:

Remarks to the Author:

Li and colleagues detail an exciting approach to locally, spatially and temporally control nozzle temperature in modified Fused Deposition Modeling (FDM) 3D printers to guide control of local printing properties. By tuning the in situ thermal gradients, the authors have demonstrated the ability to print a wide variety of materials with the same printer nozzle with Joule heating capabilities.

This work is technically interesting, but it remains to be seen the impact that this will have on the field and how and whether these phenomena are achievable as the technology scales. Achieving materials properties that are consistent for driving applications is critical for the next generation of printers and printed materials.

Reducing bubbling and degassing could have impact on the field if this were scaled appropriately. This seemed to be an aside and an afterthought in the manuscript (or perhaps a secondary point) and could turn out to be more useful to the field than the primary point of the ISHH techniques matching cast materials properties (but with the known limits of FDM printers of speed, resolution, supports, overhangs and cost).

While the samples and specimens that were printed and created are interesting, it is unclear why they need to be 3D printed and what advantages 3D printing has over other forms of thermoset manufacturing for the specific applications. During the revision process, I encourage the authors to be bold to describe and define the solution space for which ISHH allows them to achieve things not possible by other known techniques. It will be this convergence that separates out the 3D printing technologies that have a large impact on the field: ones that enable a series of structure-property relationships by processing that 1.) are not achievable by other means and 2.) align to application-specific needs. The authors must make it even more clear why ISHH is critical to achieve something specific other technologies cannot.

The figures are well constructed and useful. The results and discussion support the work well. The methodology is sound. The data support the conclusions. The missing point is clearly articulating the value of ISHH over other techniques. Figure 3 comes close to this point in comparing with cast Silicones with S8 adding in a point about surface roughness, but these are anecdotal rather than fundamental. A bit deeper discussion of crosslinking chemistry (lines 178-180) would be useful. However, the authors may open up a host of unknown and unknowable (without much deeper inquiry) issues in doing so. With chemistries that balance crosslinking and scission, it may be much more difficult to yield a controllable, consistent process.

Reviewer #2:

Remarks to the Author:

In the paper, Sun et al presented an interesting method of using direct-ink-write to 3D print structures without modifying rheological properties of the ink. The key concept is to attach a heater near the nozzle so that the extruded ink can be cured rapidly at high temperature.

- In introduction, the authors mentioned that light-curing (SLA, DLP, inkjet) is not suitable for thermosetting materials. This is incorrect. Light curing polymers are crosslinking polymers and are thus thermosets.
- Also, the statement "these technologies for complex heterogeneous designs are limited" is inaccurate. First, DLP and IJ can print much complex structures than DIW. Second, recent advances in DLP can print multimaterial structures (Science Advances, 2019, 5(5), eaav5790).
- There are also some hybrid strategies that combine DIW with DLP to achieve complicated structures, such as Peng et al, Advanced Materials, 2022, 2204890; Peng, et al, Additive Manufacturing, 2021,

40, 101911.

- Can the authors verify the degree of cure (DoC) for the printed inks?
- The curing temperature used in this paper was quite high. For printing at such high temperature (200C or above), would polymers oxidize?
- For the epoxy ink, how long did it take to cure? What was the temperature used?
- As shown in Fig 2e, the printed filaments have different aspect ratios as the temperatures in the heater and the base are changed. The temperature in the base has more influence on the aspect ratio. Were these experiments conducted on the first layer filament? Also, as the layer number increases, the printed filament becomes farther away. Would the aspect ratio change? If so, how did the author control the print height? In addition, how degree of cure is affected by these two temperatures? How does DoC change as the number of layer increases? Would mechanical properties change along the height direction?
- For the multimaterial printing, did the authors use multiple printheads, or change inks after each part was printed? Also, how are magnetizations aligned?
- The demonstration in Fig. 5s is not very clear. A top view might be helpful.

Reviewer #3:

Remarks to the Author:

In this manuscript, the authors developed a thermo-coupled 3D printing nozzle for direct ink writing. They demonstrated that this printing method is particularly good to fast-cure thermosets, and named the method in-situ hybrid heating (ISHH) printing.

First of all, the authors should revise their introduction. Some of the statements are incorrect and misleading. DLP and SLA are all producing thermosets, so telling the readers the incorrect message is not so good. The ISHH method developed by this research team is good for fast curing inks for DIW. It is an interesting technology and has its novelty (e.g. printing low viscosity inks, enhancing printing speed for DIW and removing post-curing step). However, this reviewer was a little disappointed that the team phrased it in a way that their method can solve DLP or SLA problems, which is not the case. Second, the statement of heterogenous component is not available is inconsistent with the discussion of ref 26. There is plenty of heterogenous components mentioned in that reference.

Third, the authors do not tailor the rheological properties of the ink, instead they accelerated a chemical crosslinking reaction. This strategy enabled 3D printing of a range of thermoset materials with diverse rheological properties, from low-viscosity Newtonian, shear-thinning, to yield-stress fluids. The engineering work demonstrated here is great. However, this reviewer strongly recommends that the team they should run their statements with an experienced chemist.

Complex structures and multi-material printing were achieved. The soft actuators combined with magnetic particles were printed and fabricated through this method. Also, complex geometries and structures were printed with high resolution, good self-supporting ability. In short, the reported ISHH 3D printing has offered a design-oriented 3D printing platform for many silicone-based functional materials. Therefore, the reviewer would recommend the work for a major revision.

1. Why did the Joule heater's extruded filament heated by the Joule heater have a temperature difference of 3 degrees compared with the cured silicone base instead of a more significant temperature difference?
2. Other than DSC measurement and computational modeling of the crosslinking process, were there any molecular evidence to prove the crosslinking kinetics?
3. When achieving the controllable low-viscosity and flowable window through in-situ heating, how did the authors prevent the horizontal spreading issue during 3D printing?
4. Experimental information such as instrument, experiment methods, and reagents are not provided in the manuscript or the supporting information.
5. The difference of the composition of the ink of part A and part B is not given.
6. Seven inks are used in 3D printing in Fig 4 according to the manuscript and the printed demo, but only five inks' rheology are shown in Fig. 4a&b. In addition, the labels in Fig 4a&b of inks are not

clarified in the manuscript, which makes this reviewer confused.

7. The low resolution of Fig 2 & Fig 3 makes some key information in these Figures confusing.

8. Equation 1 has different names throughout this article, including "(1)" (near where the equation itself), "Eq. 1", "Equation 1" and "equation (1)".

Responses to Comments on “NCOMMS-22-31470”

We sincerely thank three reviewers for thoroughly reading our manuscript. Your constructive comments and suggestions are fully addressed point-by-point in this letter. For your convenience, the corresponding modifications are marked in blue in the revised manuscript. We believe that the concerns raised by the editor and all reviewers have highlighted areas in need of further attention and the modifications have contributed to a significant improvement of our manuscript, making it more suitable for publication in *Nature Communications*.

Reviewer # 1

General Comment

Li and colleagues detail an exciting approach to locally, spatially and temporally control nozzle temperature in modified Fused Deposition Modeling (FDM) 3D printers to guide control of local printing properties. By tuning the in situ thermal gradients, the authors have demonstrated the ability to print a wide variety of materials with the same printer nozzle with Joule heating capabilities.

Our response

We deeply thank you for evaluating our manuscript. Note that we have changed the term “*in situ* hybrid heating (ISHH)” to “*in situ* dual heating (ISDH)” in the revised manuscript. You can refer to our **Responses 7** to Reviewer #2 for detailed description. For your convenience, we split your comments into three parts and provide our responses to each of them.

Comment #1

This work is technically interesting, but it remains to be seen the impact that this will have on the field and how and whether these phenomena are achievable as the technology scales. Achieving materials properties that are consistent for driving applications is critical for the next generation of printers and printed materials. Reducing bubbling and degassing could have impact on the field if this were scaled appropriately. This seemed to be an aside and an afterthought in the manuscript (or perhaps a secondary point) and could turn out to be more useful to the field than the primary point of the ISHH techniques matching cast materials properties (but with the known limits of FDM printers of speed, resolution, supports, overhangs and cost).

Response #1

(1) The principle of ISDH technique is heating-accelerated *in situ* gelation of the thermosetting ink. It does not require any rheology modifiers or restrain the size of the printing nozzle. The minimal nozzle diameter used in our work is 25 μm and the printing resolution is 50 μm (revised Figure 3d). To demonstrate the scalability of our ISDH technique, we added a demonstration of printing a wavy ring (diameter =90 cm) using a nozzle with 1-mm diameter. Results are now presented in the revised Fig. 3c.

On Page 9, we add:

DIW of low-viscosity thermosets

“In the absence of rheology modifiers or auxiliary reactions (except for crosslinking), the ISDH-enabled DIW is scalable when the printing nozzle varies from 25 μm to 1 mm. For example, a wrinkled ring with a diameter

of 90 mm can be successfully printed using a 1-mm nozzle (Fig. 3c) and a filament with a width of 50 μm can be also realized using a nozzle with a 25 μm inner diameter (Fig. 3d).”

Fig. 3. Characterization of ISDH-enabled DIW of low-viscosity Sylgard 184 with base-to-curing agent ratio of 10:1. **a** Photograph of printing a tube. **b** Optical and scanning electron microscope (SEM) images of a printed cylinder tube with a height of 120 mm and thickness of 1.5 mm. (Left scale bar=12 mm, right scale bar=0.8 mm). **c** Optical image of a printed wrinkled ring with a diameter of 90 mm using a nozzle with 1-mm diameter (scale bar=2 mm). **d** Serpentine network printed with a 25 μm nozzle (scale bar=100 μm).

(2) We also highlight the advantage of ISDH printing in reducing bubbles. We add a CFD simulation of 50 $^{\circ}\text{C}$ in the revised Fig. 2h and show that degassing time is significant reduced for ISDH printing. In addition, we show that even for the 1-mm nozzle, the bubble is also reduced for ISDH printing as validated by the experiment (Supplementary Fig. 5)

Revised Fig. 2h. Simulated degassing time for printing at 50, 100, 190 (ISDH) and 220 $^{\circ}\text{C}$.

Supplementary Fig. 6. Optical microscope image of printed Sylgard 184 under different conditions. **a-c** Nozzle diameter is 300 μm at **(a)** 100 $^{\circ}\text{C}$; **(b)** 190 $^{\circ}\text{C}$; **(c)** 220 $^{\circ}\text{C}$; **d-f** Nozzle diameter is 1 mm at **(a)** 100 $^{\circ}\text{C}$; **(b)** 190 $^{\circ}\text{C}$; **(c)** 220 $^{\circ}\text{C}$. (Scale bar=300 μm).

On Page 9, we add:

“When T_{heater} or T_i is too low, on the one hand, the extrude ink spreads and cannot form a filament of $h/d > 0.05$, which is undesired for DIW-based 3D printing. On the other hand, when T_{heater} or T_i is too high (e.g., 220 $^{\circ}\text{C}$), air may be trapped between filaments because the gelation time is too short for the bubble to escape (Fig. 2g and Supplementary Fig. 5). Computational fluid dynamics (CFD) simulation indicates that it takes a long time (17s) for a bubble at the bottom of the filament to escape at a temperature of 50 $^{\circ}\text{C}$, while the bubble can quickly escape when the temperature is 190 $^{\circ}\text{C}$ before the thermosetting ink fully cures (Supplementary Fig. 5). However, if the temperature gets even higher (e.g., 220 $^{\circ}\text{C}$ in Fig. 2h), the ink cures too fast so that the bubble is trapped inside. To validate the CFD results, we choose two nozzles with diameter of 300 μm and 1 mm. It is found that no bubbles are observed at 190 $^{\circ}\text{C}$ while bubbles are trapped at 220 $^{\circ}\text{C}$ for both nozzles (Supplementary Fig. 6).”

Comment #2

While the samples and specimens that were printed and created are interesting, it is unclear why they need to be 3D printed and what advantages 3D printing has over other forms of thermoset manufacturing for the specific applications. During the revision process, I encourage the authors to be bold to describe and define the solution space for which ISHH allows them to achieve things not possible by other known techniques. It will be this convergence that separates out the 3D printing technologies that have a large impact on the field: ones that enable a series of structure-property relationships by processing that 1.) are not achievable by other means and 2.) align to application-specific needs. The authors must make it even more clear why ISHH is critical to achieve something specific other technologies cannot.

Response #2

We sincerely thank your suggestions. In the revised Introduction, we clearly state the limitations of current manufacturing approaches for thermosets: “a general strategy for the fast 3D printing of diverse thermosets

with untailed material properties, high structural integrity and complexity, as well as heterogeneity and multifunctionality has not been reported yet, to the best of our knowledge”. Here, our work possesses three major novelties: (1) ISDH-enabled fast gelation that allows for direct ink writing (DIW) of thermosets (especially low-viscosity ink) with high structural complexity and low manufacturing time. (2) Diverse printable thermosets without rheological modifiers to achieve enhanced structural integrity and untailed material properties. (3) High capability of printing heterogeneous and multifunctional thermosets for emerging fields such as soft robotics and flexible electronics.

Specifically, (1) we added **Supplementary Table S1** to show the advantages of our work over conventional mold-based manufacturing approaches of thermosets; (2) We updated **Supplementary Table 2** to show the advantages of our work over existing 3D printing technologies of thermosets; (3) We revised **Abstract** and **Introduction** and articulated the limitation of existing manufacturing approaches and novelties of our work; (4) We add two more demonstrations (a magnetic stent in revised **Fig. 5d** and a flexible conductive coil in revised **Fig. 6a**) to demonstrate the capability and potential application of our ISDH-enabled DIW technique. Detailed modifications are provided below.

Supplementary Table 1. Comparison between ISDH 3D printing and existing manufacturing approaches of thermosets

Approach	Material Requirements	Manufacturing cycle time*	Geometric complexity	Facilities	Cost
Casting ³	Low viscosity resin	Days	Low	Oven, mold	Intermediate
Compression molding ¹¹	Intermediate semi-cured composite	Weeks	Low	Hydraulic press, autoclave, mold	Expensive
Reaction injection molding ^{3,11}	Low viscosity resin	Weeks	Low	Injection molding machine, mold	Expensive
ISDH 3D printing (This work)	Diverse rheological properties	Hours	High	Modified from commercial FDM 3D printers	Low cost

*Time includes the preparation (e.g., model/mold design), fabrication, and post-processing (e.g., demolding, dissolving bath-support)

Supplementary Table 2. Comparison of different approaches for 3D printing of thermosets

Type	Mechanism	Minimum nozzle /resolution (μm)	Printable height	Material applicability	Untailored thermosets property	Heterogeneous and multi-functionality	Hybrid printing
Vat polymerization	Direct sound printing by sonochemical reaction ¹⁸	None/280	High	Limited inks with specific porosity	Yes	No	No
	DLP or SLA by light-curing ¹⁵⁻¹⁷	None/20	High	Light curable resin	Yes	Yes	Yes
Direct ink writing	Rheology modification by adding particles ²³⁻²⁹	50/50	Low	Yield-stress	No	Yes	Limited
	Embedded printing in bath support ³¹⁻³⁴	100/100	High	Newtonian, shearing-thinning, yield-stress	Yes	Limited	No
	Frontal polymerization by chemical reaction ³⁶⁻³⁸	250/250	High	Limited to dicyclopentadiene	No	No	Limited
	In situ gelation by ISDH (This work)	25/50	High	Newtonian, shearing-thinning, yield-stress	Yes	Yes	Yes

Revised Abstract

Thermosets such as silicone are ubiquitous. However, existing manufacturing of thermosets involves either a prolonged manufacturing cycle (e.g., reaction injection molding), low geometric complexity (e.g., casting), or limited processable materials (e.g., frontal polymerization). Here, we report an *in situ* dual heating (ISDH) strategy for the rapid 3D printing of thermosets with complex structures and diverse rheological properties by incorporating direct ink writing (DIW) technique and a heating-accelerated *in situ* gelation mechanism. Enabled by an integrated Joule heater at the printhead, extruded thermosetting inks can quickly cure *in situ*, allowing for DIW of various thermosets with viscosities spanning five orders of magnitude, printed height over 100 mm, and high resolution of 50 μm . We further demonstrate DIW of a set of heterogeneous thermosets using multiple functional materials and present a hybrid printing of a multi-layered soft electronic circuit. Our ISDH strategy paves the way for fast manufacturing of thermosets for various emerging fields.

Revised Introduction

“Thermosets such as silicone-/epoxy-based plastics have been abundantly used in engineering, infrastructure, and daily life in the past century due to their excellent mechanical strength, thermal stability, and chemical resistance¹⁻³. Recent advances in emerging fields such as soft robotics and flexible electronics also call for the application of thermosetting components with intricate structures and multi-functionality⁴⁻⁸. To date, the manufacturing of thermosets still heavily relies on mold-based methods that allow the material to cure into its hardened forms in autoclaves or ovens, e.g., casting^{3,8-10}, compression molding¹, and reaction injection molding³. However, these conventional methods usually involve long manufacturing cycle time, low geometric complexity, and cost-prohibitive facilities, which are not amenable to fast prototyping of intricate components and have low efficacy in creating heterogeneous and multi-functional devices (Supplementary Table 1).

Recently, additive manufacturing or 3D printing has been emerging as a prominent technology for the rapid manufacturing of thermosets¹¹⁻¹³. Many strategies have been proposed to realize the 3D printing of thermosets (Supplementary Table 2). Some are utilizing vat polymerization-based 3D printing such as digital light process (DLP)¹⁴⁻¹⁶, stereolithography (SLA)¹⁷, and direct sound printing (DSP)¹⁸. For example, Kuang *et al* reported a single-vat grayscale-DLP method and a two-stage curing ink to obtain functionally graded thermosetting materials¹⁵. However, DLP and SLA are often limited to light-curable resins and DSP can only print a certain selection of thermosetting inks with specific porosity (by generating sonochemical reactions in the cavitation bubbles). Other approaches for 3D printing thermosets are based on the direct ink writing (DIW) technique¹⁹⁻²². One of the main challenges for DIW of low-viscosity thermosets is to form supporting layers to prevent structural collapse. To address this challenge, a widely used solution is to change the low-viscosity thermosetting ink to a yield-stress fluid by adding a rheology modifier such as microsized NaCl particles²³, nanosized silica²⁴⁻²⁸, and wax²⁹. However, the rheological modification inevitably tailors the material properties of the thermoset and requires modifier optimization (e.g., particle size, composition, weight fraction) for each thermoset to be successfully printed. In addition, despite the enhanced yield strength, post-curing of thermosetting polymer is usually necessary so that the printed architectures may still have relatively low heights to avoid collapse^{20-22,30}. The second strategy for DIW of thermosets is embedded 3D printing that carries out the printing process in a supporting medium³¹⁻³⁴. This method, although it can print complex structures such as silicone scaffolds³¹, often yields low structural integrity (e.g., weak filament-to-filament bonding) and has limitations in printing heterogeneous and multifunctional thermosets^{34,35}. Recently, frontal polymerization-assisted DIW of thermosets has also been reported³⁶⁻³⁸. By triggering the ring-opening metathesis polymerization (ROMP) with a catalyst, the extruded ink undergoes *in situ* gelation process at the nozzle front, allowing for quick solidification of printed thermosets. However, this ROMP-enabled *in situ* gelation mechanism seems to have been exclusively adopted

for manufacturing dicyclopentadiene-based thermosetting composites³⁸, which does not apply to other types of thermosets. Therefore, a general strategy for the fast 3D printing of diverse thermosets with untailed material properties, high structural integrity and complexity, as well as heterogeneity and multifunctionality has not been reported yet, to the best of our knowledge.

Here, we report an *in situ* dual heating (ISDH) strategy to achieve rapid 3D printing of thermosets by harnessing the heating-accelerated *in situ* gelation mechanism (Fig. 1a). By integrating a Joule heater into a printhead, freshly extruded thermosetting inks undergoes quick *in situ* gelation at an elevated temperature (Fig. 1b), allowing for fast curing of thermosets (shown by the steep increase of the viscosity in Fig. 1c). The ISDH-enabled fast gelation enables DIW of thermosets with a diverse selection of rheological and functional properties (Fig. 1d). We first demonstrate the DIW of a set of complex structures using various thermosetting inks and achieve a maximum height of 120 mm and resolution of 50 μm (Fig. 1e). In the absence of rheology modifiers or auxiliary chemical reactions (except for crosslinking), the printed thermosets preserve identical mechanical properties and chemical characteristics to the cured counterparts by the conventional molding method. We further demonstrate DIW of several heterogeneous thermosets using multiple materials (e.g., magnetic composites with different particle concentrations) and present a hybrid printing of a multilayer soft electronic circuit.”

Newly added demonstrations

Revised Fig. 5. DIW of heterogeneous and functional thermosets. e A magnetic stent printed with 20 vol% magnetic composites. The magnetizations are in the hoop direction as highlighted by black arrows. Simulated deformed shapes agree well with experiments when the magnetic field of 50 mT is applied (scale bar=8 mm).

On Page 12, we add:

“Third, the ISDH-enabled DIW can also print functional thermosets with intricate 3D structures. For example, magnetic stents that have promising applications in cardiovascular diseases have been reported in some recent works^{41,42}. Due to the hollow thin wall and highly tilted connecting ribbons, conventional molding and DIW approaches have difficulty or low efficacy in their manufacturing. Distinctly, we can directly print the magnetic stent using our ISDH-enabled DIW technology as shown in Fig. 5e. The printing process is given in Supplementary Movie 3. When a magnetic field is applied, it can have lateral shrinkage which agrees well with FEA results.”

Revised Fig. 6. Demonstration of hybrid 3D printing of soft electronics. a Schematic illustration of hybrid 3D printing of a flexible conductive coil. b Image of the flexible conductive coil. c The LED light is activated by an

alternating magnetic field. **d** The LED light keeps on when the Ecoflex substrate is deformed.

On Page 12, we add:

“By assembling pick-and-place electronic components with 3D printing (referred to as hybrid printing of soft electronics⁴³), we present two demonstrations of hybrid printing of soft electronics. First, we print a conductive coil with pick-and-place LED light on a soft Ecoflex substrate (Fig. 6a, b). The LED light can be tuned on by remotely applying an alternating magnetic field (Fig. 6c), even when the soft Ecoflex substrate is deformed (Fig. 6d).”

Comment #3

The figures are well constructed and useful. The results and discussion support the work well. The methodology is sound. The data support the conclusions. The missing point is clearly articulating the value of ISHH over other techniques. Figure 3 comes close to this point in comparing with cast Silicones with S8 adding in a point about surface roughness, but these are anecdotal rather than fundamental. A bit deeper discussion of crosslinking chemistry (lines 178-180) would be useful. However, the authors may open up a host of unknown and unknowable (without much deeper inquiry) issues in doing so. With chemistries that balance crosslinking and scission, it may be much more difficult to yield a controllable, consistent process.

Response #3

We thank the reviewer for recognizing our figures. We have provided responses to your comment on “the value of ISHH over other techniques” in **Response 2**.

Following responses are regarding the chemistry associated in our work. First, the crosslinking kinetics of thermosetting inks can be described by the autocatalytic model, which is provided in **Supplementary Text**. The gelation time is significantly reduced when the temperature is high (see **Fig. 1b**). Second, we added **density functional theory (DFT) simulations of crosslinking of Sylgard 184**. Results are provided in **Supplementary Fig. 3** (see Materials and Method for details) and show that energy barrier for crosslinking (ΔE) at 190°C is significantly reduced compared with that at 25°C. Third, we have double-checked this heating-accelerated *in situ* gelation mechanism with Prof. Ji Liu (Southern University of Science and Technology, Doctoral degree in chemistry, Expert in 3D printing of polymers). Since ISDH-printed samples have identical mechanical properties and infrared/Raman spectroscopy results to the molded counterpart at 25°C (see **Fig. 3e-h**), we are confident that heating is indeed the dominate reason. To verify this point, we intentionally made an oxidized sample and observed its different mechanical properties and infrared spectroscopy results (**Supplementary Fig. 12**).

Supplementary Fig. 3. DFT simulation of crosslinking of Sylgard 184 in which $(-\text{Si}-\text{CH}=\text{CH}_2 + \text{H}-\text{Si}- \rightarrow -\text{Si}-\text{CH}_2-\text{CH}_2-\text{Si}-)$. The energy barrier (ΔE) at 190°C (red curve) is significantly reduced compared with that at 25°C (black curve).

Supplementary Fig. 12. Comparison of ISDH printed Sylgard 184 and counterparts oxidized at 250°C for 18 hours. **a** and **b** Infrared spectroscopy of ISDH printed and oxidized samples, result shows existence of C=O band and lower CH₃, Si-O-Si and Si-C absorbance of the oxidized sample. **c** and **d** mechanical properties of ISDH printed and oxidized samples, result shows lower elastic moduli and ultimate strength of oxidized samples.

On Page 6, we add:

“The fast gelation can be further reflected by the significantly reduced energy barrier for crosslinking reactions at 190°C according to density functional theory (DFT) simulation (Supplementary Fig. 3)”

On Page 9, we add:

“It is worth noting the ISDH-printed samples have almost the same mechanical properties as the counterparts manufactured by convention molding method.”

On Page 10, we add:

“Notably, Figs. 3e-h and Supplementary Fig. 11 also suggest that no oxidization occurs during the ISDH-enabled 3D printing at 190°C. To further validate this point, we intentionally made an oxidized sample of Sylgard 184 (cured temperature is 250°C) and found that it has lower mechanical properties and different infrared spectroscopy results (e.g., a peak at 1650 cm⁻¹ shows a C=O group) (Supplementary Fig. 12).”

Reviewer # 2

General comments.

In the paper, Sun et al presented an interesting method of using direct-ink-write to 3D print structures without modifying rheological properties of the ink. The key concept is to attach a heater near the nozzle so that the extruded ink can be cured rapidly at high temperature.

Our response

We thank you for reading and commenting on our work.

Comment #1

In introduction, the authors mentioned that light-curing (SLA, DLP, inkjet) is not suitable for thermosetting materials. This is incorrect. Light curing polymers are crosslinking polymers and are thus thermosets. Also, the statement “these technologies for complex heterogeneous designs are limited” is inaccurate. First, DLP and IJ can print much complex structures than DIW. Second, recent advances in DLP can print multimaterial structures (Science Advances, 2019, 5(5), eaav5790).

Response #1

We are sorry for the inappropriate statements. We have rewritten **Introduction** and discussed the suggested reference in the revised manuscript.

On Page 3, we add:

“Recently, additive manufacturing or 3D printing has been emerging as a prominent technology for the rapid manufacturing of thermosets¹¹⁻¹³. Many strategies have been proposed to realize the 3D printing of thermosets (Supplementary Table 2). Some are utilizing vat polymerization-based 3D printing such as digital light process (DLP)¹⁴⁻¹⁶, stereolithography (SLA)¹⁷, and direct sound printing (DSP)¹⁸. For example, Kuang *et al* reported a single-vat grayscale-DLP method and a two-stage curing ink to obtain functionally graded thermosetting materials¹⁵.”

Comment #2

There are also some hybrid strategies that combine DIW with DLP to achieve complicated structures, such as Peng et al, Advanced Materials, 2022, 2204890; Peng, et al, Additive Manufacturing, 2021, 40, 101911.

Response #2

Thank you for your suggestion. These works are now cited and discussed in the revised manuscript.

On Page 13, we add:

“Inspiring works are from Qi and colleagues who have demonstrated 3D printing of complicated structures by combining DIW and DLP techniques^{44,45}”

Comment #3

Can the authors verify the degree of cure (DoC) for the printed inks?

Response #3

We use DSC measurement to verify the degree of cure for the printed inks of Sylgard 184.

On Page 9, we add:

“The degree of cure of cured Sylgard 184 is measured as 99.4% according to differential scanning calorimetry (DSC, Supplementary Fig. 7).”

Supplementary Fig. 7. DSC curve of uncured and ISDH-printed Sylgard 184. The degree of cure is calculated as 99.4%.

Comment #4

The curing temperature used in this paper was quite high. For printing at such high temperature (200°C or above), would polymers oxidize?

Response #4

This is a great question. We have to clarify that, although Joule heater is 260°C, the ink temperature is lower than 200°C in our work. Silicone-based thermosets usually do not oxidize at 200°C. We have shown that printed dogbone sample at 190°C has the same mechanical properties and infrared/Raman spectroscopy results as that of the molded counterpart at 25°C (**Fig. 3e-h**). To further validate this, we intentionally printed an oxidized sample at 250°C (e.g., a peak at 1650 cm⁻¹ shows a C=O group) and it shows different mechanical properties and infrared/Raman spectroscopy results.

On Page 10, we add:

“The ISDH strategy can also directly print Sylgard 184 with different ratios of Part A to Part B (e.g., 2:1, 5:1, 20:1) and yields molding-comparable quality. By comparing the infrared and Raman spectroscopy in Supplementary Fig. 11, we show that all printed samples have the same chemical characteristics as that of the molded counterpart. Particularly, for the 2:1 Sylgard 184, a peak at 2164.6 cm⁻¹ is observed for both ISDH-printed and molded samples, which corresponds to the Si-H stretch, indicating the existence of excessive and uncrosslinked monomers. Notably, Figs. 3e-h and Supplementary Fig. 11 also suggest that no oxidization occurs during the ISDH-enabled 3D printing at 190°C. To further validate this point, we intentionally made an oxidized sample of Sylgard 184 (cured temperature is 250°C) and found that it has lower mechanical properties and different infrared spectroscopy results (e.g., a peak at 1650 cm⁻¹ shows a C=O group) (Supplementary Fig. 12).”

Supplementary Fig. 12. Comparison of ISDH printed Sylgard 184 and oxidized samples at 250°C. a and b Infrared spectroscopy of ISDH printed and oxidized samples. Result shows existence of C=O band and lower CH₃, Si-O-Si and Si-C absorbance in the oxidized sample. c and d mechanical properties of ISDH printed and oxidized samples, result shows lower elastic moduli and ultimate strength of oxidized samples.

Comment #5

For the epoxy ink, how long did it take to cure? What was the temperature used?

Response #5

For epoxy, the temperature of Joule heater is 220°C and corresponding top surface of the cured ink base is 170°C (Fig. 2d). It takes about 1s for epoxy to cure according to the DSC measurement.

On Page 11, we add:

“For example, epoxy has an almost constant viscosity of 230 Pa·s (Supplementary Fig. 13) and it quickly cures within 1s upon heating ($T_{\text{heater}} = 240^{\circ}\text{C}$, $T_i = 170^{\circ}\text{C}$) according to DSC measurement (Supplementary Fig. 14).

Supplementary Fig. 13. Rheological properties of Sylgard186, Dragonskin 30, and epoxy. **a** Viscosity under different shear rates. **b** Storage and loss moduli under different shear stresses.

Supplementary Fig. 14. DSC curve of epoxy under 170°C. The blue area marks exothermal of crosslinking reaction. Results show the crosslinking of epoxy finishes within 1s after heating from room temperature to 170°C.

Comment #6

As shown in Fig 2e, the printed filaments have different aspect ratios as the temperatures in the heater and the base are changed. The temperature in the base has more influence on the aspect ratio. Were these experiments conducted on the first layer filament? Also, as the layer number increases, the printed filament becomes farther away. Would the aspect ratio change? If so, how did the author control the print height? In addition, how degree of cure is affected by these two temperatures? How does DoC change as the number of layer increases? Would mechanical properties change along the height direction?

Response #6

We are sorry for the misunderstanding caused. The printing platform is not a heating source. It has room temperature of 25°C and it only affects the first layer of filament. When the first layer is deposited onto the cool platform, the ink will spread a bit (as shown in Fig. 3b) because of low temperature of the printing platform. After the first layer is cured, its temperature will increase dramatically due to the heating from the Joule heater. Therefore, for any other layers, the top surface of the cured ink base will have a high temperature and serve as

the supporting base for the next-layer printing, regardless of the printing height. For example, the top surface of the cured ink base is 190 °C when $T_{\text{heater}}=260$ °C as shown in **Fig. 2e**. For a freshly extruded filament, its upper side is heated by the Joule heater and its lower sides are simultaneously heated by the top surface of the cured ink base (not by the printing platform), which is referred to as *in situ* dual heating (ISDH) in the revised manuscript. Therefore, for a given T_{heater} , the printed filaments will have the same aspect ratio, mechanical properties, and DoC except for the first layer. Results in the revised **Fig. 2** now are based on layer number >10.

When the Joule heater is removed, the temperature of the top surface of the cured ink base will also decrease (**Fig. 2f**). Therefore, when next layer is printed after a time interval, its aspect ratio will change. By tuning the time interval, we can adjust the temperature of the top surface of the cured ink base and measured the aspect ratio of the printed filament (**Fig. 2g**). The main purpose of **Fig. 2g** is to show that continuously heating from Joule heater is good to achieved high aspect ratio of filament (e.g., $h/d>0.2$, green pixels in **Fig. 2g**).

In the revised manuscript, we have changed the term “*in situ* hybrid heating (ISHH)” to “*in situ* dual heating (ISDH)” and revised **Fig. 2a, b** to elucidate the ISDH mechanism. The following sentences are added to clearly explain the ISDH-enabled 3D printing.

On Page 5, we add:

ISDH-enabled fast gelation

The underlying principle of ISDH-enabled DIW technology is the heating-accelerated gelation of the thermosetting ink. The schematic illustration of the 3D printing platform is shown in Fig. 1a and Supplementary Fig. 1 in which a noncontact Joule heater is introduced at the printhead to create a high-temperature field. The introduction of the Joule heater has a twofold effect. First, it continuously heats the upper side of the freshly extruded ink; Second, it keeps the top surface of the already cured ink base at a high temperature. Therefore, when the ink is extruded and deposited onto the cured ink base, its upper and lower sides are simultaneously heated *in situ*, which is referred to as *in situ* dual heating. The thermosetting ink remains at a low temperature in the nozzle by employing a heat isolator and a fluidic cooling system. After being extruded and deposited, it undergoes a fast *in situ* gelation process due to the drastic increase in temperature, and thereafter serves as the support for printing the next layer.

Fig. 2. The mechanism of *in situ* dual heating. **a** Schematic illustration of ISDH. The freshly extruded ink is labeled as $(i+1)^{\text{th}}$ layer and the top layer of the cured ink base is labeled as i^{th} layer. The temperature of the Joule heater, $(i+1)^{\text{th}}$ layer, and i^{th} layer is denoted as T_{heater} , T_{i+1} , and T_i , respectively. **b** Simulated degassing time at 50, 100, 190 and 220 °C. **c** Experimental and simulated temperature of freshly extruded ink when $i = 2, 5, 10$. **d** Finite element analysis of the thermal flux on the upper and lower side of the freshly extruded ink. **e** T_i as a function of T_{heater} during continuous heating. **f** Temperature of i^{th} layer as a function of time after the Joule heater is removed. **g** The aspect ratio of the filament's cross-section h/d is plotted as a function of T_{heater} and T_i using Sylgard 184. The unreachable domain refers to T_i greater than the corresponding value in (e) under the same T_{heater} .

On Page 6, we add:

“To elucidate the ISDH mechanism, we label the freshly extruded ink/filament and the top surface of the cured ink base as $(i + 1)^{\text{th}}$ and i^{th} layer, respectively (Fig. 2a). The temperature of Joule heater, $(i + 1)^{\text{th}}$ layer, and i^{th} layer is denoted as T_{heater} , T_{i+1} , and T_i , respectively. When the ink is extruded and deposited on the cured ink base, its upper side is heated by the Joule heater via convection and radiation, while its lower side is heated by the i^{th} layer via conduction. The temperature of the $(i + 1)^{\text{th}}$ and i^{th} layers are measured using an infrared camera during printing (Supplementary Fig. 4). By setting $T_{\text{heater}} = 260^{\circ}\text{C}$, the i^{th} layer can maintain an almost constant temperature $T_i = 190^{\circ}\text{C}$ regardless of the printing height (e.g., $T_i \equiv 190^{\circ}\text{C}$ for $i = 10, 12, 15$ when $T_{\text{heater}} = 260^{\circ}\text{C}$ in Fig. 2b). Therefore, for any freshly extruded ink, its temperature can quickly increase to $T_{i+1} = 196^{\circ}\text{C}$ at $t=2\text{s}$ after being deposited on the i^{th} layer. To further investigate the heat transfer inside the freshly extruded ink, we conduct finite element analysis (FEA) using the commercial software COMSOL Multiphysics. The temperature change and heat flux through the upper and lower side of the $(i + 1)^{\text{th}}$ layer are presented in Fig. 2c. Initially, the freshly extruded ink (i.e., $(i + 1)^{\text{th}}$ layer) has a temperature of 25°C . When deposited on the cured ink base, its lower side is quickly heated up (see the temperature contour plot at $t=0.5\text{s}$) by the i^{th} layer. During this period, the heat flux from the lower side is one order of magnitude larger than that through upper side from the Joule heater (Fig. 2d). After being heated to 190°C , heat flux through the lower side attenuates, while the freshly extruded ink is continuously heated by the Joule heater. Note that the $(i + 1)^{\text{th}}$ layer has a relatively uniform temperature distribution at $t=2\text{s}$ in which the upper side is 196°C and the lower side is 193°C (see the temperature contour plot at $t=2\text{s}$ in Fig. 2c), giving rise to consistent gelation inside the freshly extruded ink. Thereafter, the cured $(i + 1)^{\text{th}}$ layer will serve as the base for supporting next layer of freshly extruded ink.

The continuous heating from Joule heater keeps the top surface of the cured ink base at a constant temperature. For example, $T_i \equiv 190^{\circ}\text{C}$ when $T_{\text{heater}} = 260^{\circ}\text{C}$. When T_{heater} is changed, T_i will be altered accordingly (Fig. 2e) and if the Joule heater is removed, T_i will also rapidly decrease (Fig. 2f). Both T_{heater} and T_i affect the aspect ratio of the single filament's cross-section (denoted as h/d in Fig. 2a). We measured the aspect ratio h/d of printed filament at different T_{heater} and T_i values and results are presented in Fig. 2g. It shows that h/d increases when T_{heater} and T_i increases due to the faster gelation. When T_{heater} or T_i is too low, on the one hand, the extrude ink spreads and cannot form a filament of $h/d > 0.05$, which is undesired for DIW-based 3D printing. On the other hand, when T_{heater} or T_i is too high (e.g., 220°C), air may be trapped between filaments because the gelation time is too short for the bubble to escape (Fig. 2g and Supplementary Fig. 5). Computational fluid dynamics (CFD) simulation indicates that it takes a long time (17s) for a bubble at the bottom of the filament to escape at a temperature of 50°C , while the bubble can quickly escape when the temperature is 190°C before the thermosetting ink fully cures (Supplementary Fig. 5). However, if the temperature gets even higher (e.g., 220°C in Fig. 2h), the ink cures too fast so that the bubble is trapped inside. To validate the CFD results, we choose two nozzles with diameter of 300 μm and 1 mm. It is found that no bubbles are observed at 190°C while bubbles are trapped at 220°C for both nozzles. Therefore, a good choice for printing filaments with high aspect ratio ($h/d > 0.2$, green pixels in Fig. 2g) and reduced bubbles is to use $T_{\text{heater}} = 240\sim 260^{\circ}\text{C}$ (corresponding $T_i = 170\sim 190^{\circ}\text{C}$). Also, to achieve consistent structural integrity, we suggest continuous heating from Joule heating. Otherwise, reheating the cured ink base is necessary if the heating is interrupted.”

Comment #7

For the multimaterial printing, did the authors use multiple printheads, or change inks after each part was printed? Also, how are magnetizations aligned?

Response #7

We change the ink after each part is printed.

On Page 11, we add:

“Note that the ink is changed after each part is printed.” We have also revised **Fig. 5** and magnetization directions are denoted by the black arrows. We clarify this in **Materials and Methods**: After printing, the magnetic thermosets were first folded into a pre-designed shape and then magnetized by an impulse magnetic field (2.8 T) by an impulse magnetizer (M20-2020, HLT Co. LTD, China).

Fig. 5. DIW of heterogeneous and functional thermosets. **a** Heterogeneous thermosets using Sylgard 184, DragonSkin 10, and DragonSkin 10 +SiO₂ (scale bar=5 mm). **b** Heterogeneous thermosets using epoxy and Ecoflex 00-30 (scale bar=5 mm). **c** A magnetic beam with five segments in which NdFeB vol% varies from 0 to 20%. The magnetization directions are highlighted by black arrows. **d** A magnetic flower with three petals in which NdFeB vol% are 5, 15, and 20%. The magnetizations are pointing down. **e** A magnetic stent printed with 20 vol% composites. The magnetizations are in the hoop direction as highlighted by black arrows. Simulated deformed shapes agree well with experiments when the magnetic field of 50 mT is applied (scale bar=8 mm).

Comment #8

The demonstration in Fig. 5s is not very clear. A top view might be helpful.

Response #8

Fig. 5 has been revised. 3D schematics are now provided. See our **Response 7** above.

Reviewer #3

General Comments.

In this manuscript, the authors developed a thermo-coupled 3D printing nozzle for direct ink writing. They demonstrated that this printing method is particularly good to fast-cure thermosets, and named the method in-situ hybrid heating (ISHH) printing.

Our response

We thank the reviewer for reading and evaluating our work. For your convenience, we separate your comments into a few points and provide our responses to each of them.

Comment #1

First of all, the authors should revise their introduction. Some of the statements are incorrect and misleading. DLP and SLA are all producing thermosets, so telling the readers the incorrect message is not so good. The ISHH method developed by this research team is good for fast curing inks for DIW. It is an interesting technology and has its novelty (e.g. printing low viscosity inks, enhancing printing speed for DIW and removing post-curing step). However, this reviewer was a little disappointed that the team phrased it in a way that their method can solve DLP or SLA problems, which is not the case. Third, the authors do not tailor the rheological properties of the ink, instead they accelerated a chemical crosslinking reaction. This strategy enabled 3D printing of a range of thermoset materials with diverse rheological properties, from low-viscosity Newtonian, shear-thinning, to yield-stress fluids. The engineering work demonstrated here is great. However, this reviewer strongly recommends that the team they should run their statements with an experienced chemist.

Complex structures and multi-material printing were achieved. The soft actuators combined with magnetic particles were printed and fabricated through this method. Also, complex geometries and structures were printed with high resolution, good self-supporting ability. In short, the reported ISHH 3D printing has offered a design-oriented 3D printing platform for many silicone-based functional materials. Therefore, the reviewer would recommend the work for a major revision.

Response #1

We sincerely thank the reviewer for recognizing the novelty of our work. Also, we are sorry for the inappropriate statements. To strengthen our work, we have consulted Prof. Ji Liu (Southern University of Science and Technology, Doctoral degree in chemistry, Expert in 3D printing of polymers, now he is added as the co-author) and significantly revised our manuscript.

Specifically, (1) we added **Supplementary Table S1** to show the advantages of our work over conventional mold-based manufacturing approaches of thermosets; (2) We updated **Supplementary Table 2** to show the advantages of our work over existing 3D printing technologies of thermosets; (3) We revised **Abstract** and **Introduction** and articulated the limitation of existing manufacturing approaches and novelties of our work; (4) We add two more demonstrations (**a magnetic stent in revised Fig. 5d** and **a flexible conductive coil in revised Fig. 6a**) to demonstrate the capability and potential application of our ISDH-enabled DIW technique. Detailed modifications are provided below.

Supplementary Table 1. Comparison between ISDH 3D printing and existing manufacturing approaches of thermosets

Approach	Material Requirements	Manufacturing cycle time*	Geometric complexity	Facilities	Cost
Casting ³	Low viscosity resin	Days	Low	Oven, mold	Intermediate
Compression molding ¹¹	Intermediate semi-cured composite	Weeks	Low	Hydraulic press, autoclave, mold	Expensive
Reaction injection molding ^{3,11}	Low viscosity resin	Weeks	Low	Injection molding machine, mold	Expensive
ISDH 3D printing (This work)	Diverse rheological properties	Hours	High	Modified from commercial FDM 3D printers	Low cost

*Time includes the preparation (e.g., model/mold design), fabrication, and post-processing (e.g., demolding, dissolving bath-support)

Supplementary Table 2. Comparison of different approaches for 3D printing of thermosets

Type	Mechanism	Minimum nozzle /resolution (μm)	Printable height	Material applicability	Untailored thermosets property	Heterogeneous and multi-functionality	Hybrid printing
Vat polymerization	Direct sound printing by sonochemical reaction ¹⁸	None/280	High	Limited inks with specific porosity	Yes	No	No
	DLP or SLA by light-curing ¹⁵⁻¹⁷	None/20	High	Light curable resin	Yes	Yes	Yes
Direct ink writing	Rheology modification by adding particles ²³⁻²⁹	50/50	Low	Yield-stress	No	Yes	Limited
	Embedded printing in bath support ³¹⁻³⁴	100/100	High	Newtonian, shearing-thinning, yield-stress	Yes	Limited	No
	Frontal polymerization by chemical reaction ³⁶⁻³⁸	250/250	High	Limited to dicyclopentadiene	No	No	Limited
	In situ gelation by ISDH (This work)	25/50	High	Newtonian, shearing-thinning, yield-stress	Yes	Yes	Yes

Revised Abstract

Thermosets such as silicone are ubiquitous. However, existing manufacturing of thermosets involves either a prolonged manufacturing cycle (e.g., reaction injection molding), low geometric complexity (e.g., casting), or limited processable materials (e.g., frontal polymerization). Here, we report an *in situ* dual heating (ISDH) strategy for the rapid 3D printing of thermosets with complex structures and diverse rheological properties by incorporating direct ink writing (DIW) technique and a heating-accelerated *in situ* gelation mechanism. Enabled by an integrated Joule heater at the printhead, extruded thermosetting inks can quickly cure *in situ*, allowing for DIW of various thermosets with viscosities spanning five orders of magnitude, printed height over 100 mm, and high resolution of 50 μm . We further demonstrate DIW of a set of heterogeneous thermosets using multiple functional materials and present a hybrid printing of a multi-layered soft electronic circuit. Our ISDH strategy paves the way for fast manufacturing of thermosets for various emerging fields.

Revised Introduction

“Thermosets such as silicone-/epoxy-based plastics have been abundantly used in engineering, infrastructure, and daily life in the past century due to their excellent mechanical strength, thermal stability, and chemical resistance¹⁻³. Recent advances in emerging fields such as soft robotics and flexible electronics also call for the application of thermosetting components with intricate structures and multi-functionality⁴⁻⁸. To date, the manufacturing of thermosets still heavily relies on mold-based methods that allow the material to cure into its hardened forms in autoclaves or ovens, e.g., casting^{3,8-10}, compression molding¹, and reaction injection molding³. However, these conventional methods usually involve long manufacturing cycle time, low geometric complexity, and cost-prohibitive facilities, which are not amenable to fast prototyping of intricate components and have low efficacy in creating heterogeneous and multi-functional devices (Supplementary Table 1).

Recently, additive manufacturing or 3D printing has been emerging as a prominent technology for the rapid manufacturing of thermosets¹¹⁻¹³. Many strategies have been proposed to realize the 3D printing of thermosets (Supplementary Table 2). Some are utilizing vat polymerization-based 3D printing such as digital light process (DLP)¹⁴⁻¹⁶, stereolithography (SLA)¹⁷, and direct sound printing (DSP)¹⁸. For example, Kuang *et al* reported a single-vat grayscale-DLP method and a two-stage curing ink to obtain functionally graded thermosetting materials¹⁵. However, DLP and SLA are often limited to light-curable resins and DSP can only print a certain selection of thermosetting inks with specific porosity (by generating sonochemical reactions in the cavitation bubbles). Other approaches for 3D printing thermosets are based on the direct ink writing (DIW) technique¹⁹⁻²². One of the main challenges for DIW of low-viscosity thermosets is to form supporting layers to prevent structural collapse. To address this challenge, a widely used solution is to change the low-viscosity thermosetting ink to a yield-stress fluid by adding a rheology modifier such as microsized NaCl particles²³, nanosized silica²⁴⁻²⁸, and wax²⁹. However, the rheological modification inevitably tailors the material properties of the thermoset and requires modifier optimization (e.g., particle size, composition, weight fraction) for each thermoset to be successfully printed. In addition, despite the enhanced yield strength, post-curing of thermosetting polymer is usually necessary so that the printed architectures may still have relatively low heights to avoid collapse^{20-22,30}. The second strategy for DIW of thermosets is embedded 3D printing that carries out the printing process in a supporting medium³¹⁻³⁴. This method, although it can print complex structures such as silicone scaffolds³¹, often yields low structural integrity (e.g., weak filament-to-filament bonding) and has limitations in printing heterogeneous and multifunctional thermosets^{34,35}. Recently, frontal

polymerization-assisted DIW of thermosets has also been reported³⁶⁻³⁸. By triggering the ring-opening metathesis polymerization (ROMP) with a catalyst, the extruded ink undergoes *in situ* gelation process at the nozzle front, allowing for quick solidification of printed thermosets. However, this ROMP-enabled *in situ* gelation mechanism seems to have been exclusively adopted for manufacturing dicyclopentadiene-based thermosetting composites³⁸, which does not apply to other types of thermosets. Therefore, a general strategy for the fast 3D printing of diverse thermosets with untailed material properties, high structural integrity and complexity, as well as heterogeneity and multifunctionality has not been reported yet, to the best of our knowledge.

Here, we report an *in situ* dual heating (ISDH) strategy to achieve rapid 3D printing of thermosets by harnessing the heating-accelerated *in situ* gelation mechanism (Fig. 1a). By integrating a Joule heater into a printhead, freshly extruded thermosetting inks undergoes quick *in situ* gelation at an elevated temperature (Fig. 1b), allowing for fast curing of thermosets (shown by the steep increase of the viscosity in Fig. 1c). The ISDH-enabled fast gelation enables DIW of thermosets with a diverse selection of rheological and functional properties (Fig. 1d). We first demonstrate the DIW of a set of complex structures using various thermosetting inks and achieve a maximum height of 120 mm and resolution of 50 μm (Fig. 1e). In the absence of rheology modifiers or auxiliary chemical reactions (except for crosslinking), the printed thermosets preserve identical mechanical properties and chemical characteristics to the cured counterparts by the conventional molding method. We further demonstrate DIW of several heterogeneous thermosets using multiple materials (e.g., magnetic composites with different particle concentrations) and present a hybrid printing of a multilayer soft electronic circuit.”

Newly added demonstrations

Revised Fig. 5. DIW of heterogeneous and functional thermosets. e A magnetic stent printed with 20 vol% magnetic composites. The magnetizations are in the hoop direction as highlighted by black arrows. Simulated deformed shapes agree well with experiments when the magnetic field of 50 mT is applied (scale bar=8 mm).

On Page 12, we add:

“Third, the ISDH-enabled DIW can also print functional thermosets with intricate 3D structures. For example, magnetic stents that have promising applications in cardiovascular diseases have been reported in some recent works^{41,42}. Due to the hollow thin wall and highly tilted connecting ribbons, conventional molding and DIW approaches have difficulty or low efficacy in their manufacturing. Distinctly, we can directly print the magnetic stent using our ISDH-enabled DIW technology as shown in Fig. 5e. The printing process is given in Supplementary Movie 3. When a magnetic field is applied, it can have lateral shrinkage which agrees well with FEA results.”

Revised Fig. 6. Demonstration of hybrid 3D printing of soft electronics. **a** Schematic illustration of hybrid 3D printing of a flexible conductive coil. **b** Image of the flexible conductive coil. **c** The LED light is activated by an alternating magnetic field. **d** The LED light keeps on when the Ecoflex substrate is deformed.

On Page 12, we add:

“By assembling pick-and-place electronic components with 3D printing (referred to as hybrid printing of soft electronics⁴³), we present two demonstrations of hybrid printing of soft electronics. First, we print a conductive coil with pick-and-place LED light on a soft Ecoflex substrate (Fig. 6a, b). The LED light can be tuned on by remotely applying an alternating magnetic field (Fig. 6c), even when the soft Ecoflex substrate is deformed (Fig. 6d).”

Comment #2

Why did the Joule heater's extruded filament heated by the Joule heater have a temperature difference of 3 degrees compared with the cured silicone base instead of a more significant temperature difference?

Response #2

A brief answer to your question is that the heat transfer from Joule heater (radiation and convection via air) has less effect than the heat conduction from the cured ink base. As shown in Supplementary Fig. 1b, the height of the nozzle and heater with respect to the cured ink base is $H_{\text{nozzle}} = 0.3 \text{ mm}$ and $H_{\text{heater}} = 1 \text{ mm}$, respectively. The air gap between the Joule heater and freshly extruded ink is 0.7 mm . Under continuous heating from the Joule heater (260°C), the top surface of the cured ink base remains about 190°C . When the ink is extruded, the heat flux from the cured ink base has more dominating effect than the conduction from the Joule heater (see Fig. 2d) first. Thereafter, when the temperature of the freshly extruded ink reaches 190°C , heat flux from lower side attenuates, and continuous heating from Joule heater will increase its temperature by $\sim 3^\circ\text{C}$.

Supplementary Fig. 1. (b) Control parameters for ISDH printing. Key parameters including the height of the nozzle and heater with respect to the cured ink base ($H_{\text{nozzle}} = 0.3 \text{ mm}$ and $H_{\text{heater}} = 1 \text{ mm}$), the temperature of

the heater and cooler (T_{heater} and T_{cooler}), air pressure applied for ink extrusion (P) and moving speed of the nozzle ($v=30$ mm/s).

To clearly elucidate ISDH mechanism, we have revised Fig. 2 and provided both FEA and experimental measurements of the temperature for different layers of freshly extruded ink.

Fig. 2. The mechanism of *in situ* dual heating. **a** Schematic illustration of ISDH. The freshly extruded ink is labeled as $(i+1)^{\text{th}}$ layer and the top layer of the cured ink base is labeled as i^{th} layer. The temperature of the Joule heater, $(i+1)^{\text{th}}$ layer, and i^{th} layer is denoted as T_{heater} , T_{i+1} , and T_i , respectively. **b** Simulated degassing time at 50, 100, 190, and 220°C. **c** Experimental and simulated temperature of freshly extruded ink when $i = 2, 5, 10$. **d** Finite element analysis of the thermal flux on the upper and lower side of the freshly extruded ink. **e** T_i as a function of T_{heater} during continuous heating. **f** Temperature of i^{th} layer as a function of time after the Joule heater is removed. **g** The aspect ratio of the filament's cross-section h/b is plotted as a function of T_{heater} and T_i using Sylgard 184. The unreachable domain refers to T_i greater than

the corresponding value in (e) under the same T_{heater} .

On Page 7, we add:

“To elucidate the ISDH mechanism, we label the freshly extruded ink/filament and the top surface of the cured ink base as $(i + 1)^{\text{th}}$ and i^{th} layer, respectively (Fig. 2a). The temperature of Joule heater, $(i + 1)^{\text{th}}$ layer, and i^{th} layer is denoted as T_{heater} , T_{i+1} , and T_i , respectively. When the ink is extruded and deposited on the cured ink base, its upper side is heated by the Joule heater via convection and radiation, while its lower side is heated by the i^{th} layer via conduction. The temperature of the $(i + 1)^{\text{th}}$ and i^{th} layers are measured using an infrared camera during printing (Supplementary Fig. 4). By setting $T_{\text{heater}} = 260^{\circ}\text{C}$, the i^{th} layer can maintain an almost constant temperature $T_i = 190^{\circ}\text{C}$ regardless of the printing height (e.g., $T_i \equiv 190^{\circ}\text{C}$ for $i = 10, 12, 15$ when $T_{\text{heater}} = 260^{\circ}\text{C}$ in Fig. 2b). Therefore, for any freshly extruded ink, its temperature can quickly increase to $T_{i+1} = 196^{\circ}\text{C}$ at $t=2\text{s}$ after being deposited on the i^{th} layer. To further investigate the heat transfer inside the freshly extruded ink, we conduct finite element analysis (FEA) using the commercial software COMSOL Multiphysics. The temperature change and heat flux through the upper and lower side of the $(i + 1)^{\text{th}}$ layer are presented in Fig. 2c. Initially, the freshly extruded ink (i.e., $(i + 1)^{\text{th}}$ layer) has a temperature of 25°C . When deposited on the cured ink base, its lower side is quickly heated up (see the temperature contour plot at $t=0.5\text{s}$) by the i^{th} layer. During this period, the heat flux from the lower side is one order of magnitude larger than that through upper side from the Joule heater (Fig. 2d). After being heated to 190°C , heat flux through the lower side attenuates, while the freshly extruded ink is continuously heated by the Joule heater. Note that the $(i + 1)^{\text{th}}$ layer has a relatively uniform temperature distribution at $t=2\text{s}$ in which the upper side is 196°C and the lower side is 193°C (see the temperature contour plot at $t=2\text{s}$ in Fig. 2c), giving rise to consistent gelation inside the freshly extruded ink. Thereafter, the cured $(i + 1)^{\text{th}}$ layer will serve as the base for supporting next layer of freshly extruded ink.”

Comment #3

Other than DSC measurement and computational modeling of the crosslinking process, were there any molecular evidence to prove the crosslinking kinetics?

Response #3

Thank you for your suggestions. First, the crosslinking kinetics of thermosetting inks can be described by the autocatalytic model, which is provided in **Supplementary Text**. The gelation time is significantly reduced when the temperature is high (see **Fig. 1b**). Second, we added **density functional theory (DFT) simulations of crosslinking of Sylgard 184**. Results are provided in **Supplementary Fig. 3** (see **Materials and Methods** for details) and show that energy barrier for crosslinking (ΔE) at 190°C is significantly reduced compared with that at 25°C .

Supplementary Fig. 3. DFT simulation of crosslinking of Sylgard 184 in which $(-\text{Si}-\text{CH}=\text{CH}_2 + \text{H}-\text{Si}- \rightarrow -\text{Si}-\text{CH}_2-\text{CH}_2-\text{Si}-)$. The energy barrier (ΔE) at 190°C (red curve) is significantly reduced compared with that at 25°C (black curve).

On Page 6, we add:

“The fast gelation can be further reflected by the significantly reduced energy barrier for crosslinking reactions at 190°C according to density functional theory (DFT) simulation (Supplementary Fig. 3)”

Comment #4

When achieving the controllable low-viscosity and flowable window through in-situ heating, how did the authors prevent the horizontal spreading issue during 3D printing?

Response #4

The ink remains flowable fluid with low temperature ($\sim 25^\circ\text{C}$) in the nozzle by the fluidic cooling system. By controlling the temperature of the Joule heater (denoted as T_{heater}), the temperature of the top surface of the cured ink base (denoted as T_i) is accordingly controlled. When the ink is extruded and deposited on the cured ink base, it undergoes fast gelation within 1-2s, which prevents the spreading. This is the key point of our work: *in situ* dual heating-enabled fast gelation mechanism. The cross-sectional aspect ratio of printed filament (denoted as h/d in Fig. 2a) as a function of temperature of Joule heater and top surface of cured ink base is provided in Fig. 2g.

Revised Fig. 2 The mechanism of *in situ* dual heating. **a** Schematic illustration of ISDH. The freshly

extruded ink is labeled as $(i + 1)^{\text{th}}$ layer and the top layer of the cured ink base is labeled as i^{th} layer. The temperature of the Joule heater, $(i + 1)^{\text{th}}$ layer, and i^{th} layer is denoted as T_{heater} , T_{i+1} , and T_i , respectively. **g** The aspect ratio of the filament's cross-section h/b is plotted as a function of T_{heater} and T_i using Sylgard 184.

On Page 8, we add:

“Both T_{heater} and T_i affect the aspect ratio of the single filament's cross-section (denoted as h/d in Fig. 2a). We measured the aspect ratio h/d of printed filament at different T_{heater} and T_i values and results are presented in Fig. 2g. It shows that h/d increases when T_{heater} and T_i increases due to the faster gelation. When T_{heater} or T_i is too low, on the one hand, the extrude ink spreads and cannot form a filament of $h/d > 0.05$, which is undesired for DIW-based 3D printing.”

“Therefore, a good choice for printing filaments with high aspect ratio ($h/d > 0.2$, green pixels in Fig. 2g) and reduced bubbles is to use $T_{\text{heater}} = 240 \sim 260^{\circ}\text{C}$ (corresponding $T_i = 170 \sim 190^{\circ}\text{C}$).”

Comment #5

Experimental information such as instrument, experiment methods, and reagents are not provided in the manuscript or the supporting information.

Response #5

Thank you for your suggestions. We have revised **Materials and Methods** and provided detailed information with subheadings.

Materials and Methods

Ink preparation

Pure thermosetting ink. Sylgard 184 and Sylgard 186 are purchased from Dow Corning, USA. Dragonskin 30, Dragonskin 10, Ecoflex 00-30, and Ecoflex 00-10 are purchased from Smooth-on, USA. The silicone ink was prepared by directly blending corresponding Part A and B at a specific mass ratio in a 50 ml beaker. For Sylgard 184, Part A: B=10:1 is used unless other mixing ratios are specified. For 186, Part A: B=10:1. For Dragonskin 30, Dragonskin 10, Ecoflex 00-30, and Ecoflex 00-10, Part A: B=1:1. For Dragonskin 10 + 20 wt%SiO₂ and blended using a planetary mixer (ARV-310, Thinky, Japan) at 2000 rpm for 5 min. The epoxy ink is purchased from Sinoepc China, and mixing E39D and E20 with 1:1 weight ratio by manual stirring under heating with a heat gun, then added 14.2 phmr 2-ethyl-4-methylimidazole (Aladdin, China) as curing agent. For all these materials, colorants are added for identification.

Magnetic composite ink. The magnetic composite ink is prepared by adding NdFeB microparticles with an average size of 5 μm (MQFP-B+, Magnequench, Canada) into Sylgard 184 with different volume ratios and blended using a planetary mixer (ARV-310, Thinky, Japan) at 2000 rpm for 5 min.

Conductive Ink. The conductive ink is prepared by adding Ag flakes (100*2 μm , Shanghai Xinzuan Alloy Material Co., Ltd., China) and fluoroelastomer (FPM) (product number: G801, DAI-EL, Japan) to DMC (product number: 517127, anhydrous, $\geq 99\%$, Sigma–Aldrich, USA) with a weight ratio FPM: DMC: Ag = 1:7:3 and then mixed using a planetary mixer (ARV-310, Thinky, Japan) at 2000 rpm for 30 min.

ISDH 3D printing procedure

ISDH printing of pure thermosetting inks. The ISDH 3D printer is adapted from a commercial FDM printer (Ultimaker 2+ Extended, Ultimaker, Netherland). A ceramic ring-shaped Joule heater with an embedded K-type thermocouple is mounted to the printhead. A thermal isolator is covered on the heater to reduce heat dissipation. A liquid chiller is customized and fabricated by selective laser sintering (SLS) technology to fit the nozzle shape. Before printing, all the as-prepared inks were transferred into 30 cc syringe barrels (PSY-30F, MUSASHI engineering, Japan) and centrifuged at 2200 rpm for 5 time to remove bubbles. Then, the syringe barrels were connected to a digital pneumatic regulator (Super Σ CMIII, MUSASHI engineering, Japan) and mounted to the printhead. Nozzles with a 25 μm (XYMtech, China), 300 μm and 1 mm inner diameter (MUSASHI engineering, Japan) were used for demonstrations. After turning on the Joule heater and chiller for 5 minutes for thermal balance, distances between the Joule heater, nozzle tip, and base plate are carefully calibrated to $H_{\text{heater}}=1$ mm and $H_{\text{nozzle}}=0.08$ mm (Supplementary Fig. 1b). Printing paths were generated by self-designed STL models (SolidWorks, Dassault Systèmes, France) and converted into G-code by the commercial software Ultimaker Cura (Ultimaker, Netherland). Precleaned glass plates are used as platform for printing.

ISDH printing of magnetic thermosets. Magnetic composites with different NdFeB volume fraction were printed subsequently by changing inks after each part is printed. After printing, the magnetic thermosets were first folded into a predesigned shape and then magnetized by an impulse magnetic field (2.8 T) by an impulse magnetizer (M20-2020, HLT Co. LTD, China).

Hybrid printing of soft electronics. The soft electronics was designed using the commercial software Altium Designer (Altium Limited, Australia) and then transferred into printing paths. Pick-and-place electronic components are directly pick and placed at designated areas.

Characterization

Rheological properties. The rheological properties of the inks were measured by a rotational rheometer (Physical MCR 301, Anton Paar, Austrian). Before each test, the inks were kept isothermally at 25°C for 5 min. During the viscometry measurements, the shear rate was swept from 0.01 s^{-1} to 100 s^{-1} . During the oscillatory measurements, shear stress was swept from 1 Pa to 1000 Pa at 1 Hz.

DSC measurement. The crosslinking properties of the inks were measured by differential scanning calorimetry. The inks were first kept isothermally at 25°C for 5 min and then heated to 200°C at different rates under nitrogen flow (10°C/min, 20°C/min, 25°C/min, and 30°C/min).

Mechanical properties. The specimens for mechanical tests were printed or cast into a dogbone specimen with a cross-section of 2 mm \times 10 mm. Tensile properties were measured on a universal testing machine with a constant crosshead speed of 10 mm s^{-1} (AGS-X, SHIMADZU, Japan).

Infrared/Raman spectroscopy. The image of the printed filament was measured using a scanning electron microscope (Hitachi, SU8220, Japan). The surface quality of the printed filament was characterized by an atomic force microscope (Dimension Icon, Bruker, Germany). The cross-section of the printed filaments was measured using an optical microscope (DSX510, Olympus, Japan). All optical images and videos in this study were recorded using a CMOS camera (acA2440-20 gm/gc, Basler, Germany). The infrared image and video for temperature measurement were recorded using an infrared camera (ETS320, FLIR, USA). The temperature was recorded using FLIR Tools+ software.

Numerical simulations

Heat transfer. Heat transfer of the freshly extruded ink during the ISDH printing process was conducted using the commercial software COMSOL Multiphysics. The thermodynamics coefficients of Sylgard 184 and air are adopted from reference^{42,43} and are listed in Supplementary Table 3.

CFD simulations. CFD simulation for degassing was conducted using the commercial software Fluent. A user-defined function (UDF) was developed to describe the viscosity of Sylgard 184 under different conditions according to Eq. 1 with coefficients listed in Supplementary Table 4.

FEA of magnetic composites. The deformation of the NdFeB magnetic composites under a uniform magnetic field was simulated in the commercial package ABAQUS. To account for the interaction between magnetic composite with embedded NdFeB microparticles with the uniform magnetic field, we adopt a user's subroutine developed by Zhao *et al*⁴⁶. In brief, the magnetic torque density $\tau = \mathbf{M} \times \mathbf{B}$ can be implemented by computing the magnetic Cauchy stress $\sigma^{\text{magnetic}} = -\mathbf{B} \otimes \mathbf{F} \mathbf{M}$ where \mathbf{F} is the deformation gradient and operator \otimes represents a dyadic product that takes two vectors to yield a second-order tensor. The Young's modulus and magnetization of the magnetic composite with different NdFeB vol% are calculated according to Wang *et al*⁴⁷. Details are provided in the Supplementary Text.

DFT simulations. DFT models of molecules (base monomer and crosslinker) were built using GAUSSVIEW6⁴⁸. All calculations were done with GAUSSIAN09W program. The DFT method at the B3LYP/6-31G* level of theory was applied in transition states search, intrinsic reaction coordinate (IRC) calculation and geometry optimism. B3LYP/6-311G** level of theory was applied in single-point energy calculation. The transition states were obtained using QST3 method and the stationary points were confirmed by calculation of the energy second derivatives with respect to atomic coordinates. The thermodynamics were calculated using Shermo⁴⁹.

Comment #6

The difference of the composition of the ink of part A and part B is not given.

Response #6

We provided the chemical composition of Part A and B in Supplementary Fig. 2.

Supplementary Fig. 2. Chemical composition and crosslinking reaction of Sylgard 184.

On Page 6, we add:

“Take a typical thermosetting material, silicone Sylgard 184, as an example. As a two-part silicone, the mixing ratio of Part A (base monomer) and Part B (crosslinker) (compositions are shown in Supplementary Fig. 2)

will change the viscosity of Sylgard 184. By default, Sylgard 184 in this work refers to that with Part A: Part B=10:1 mass ratio unless other mixing ratios are specified.”

Comment #7

Seven inks are used in 3D printing in Fig 4 according to the manuscript and the printed demo, but only five inks' rheology are shown in Fig. 4a&b. In addition, the labels in Fig 4a&b of inks are not clarified in the manuscript, which makes this reviewer confused.

Response #7

Since we demonstrate 9 different thermosetting inks, if all inks are presented in main figure, the figure will be crowded and messy. Therefore, we present 5 representative ink (including low-viscosity Sylgard 184 and Ecoflex, yield-stress Dragonskin) in **Fig. 4a,b** and others in **Supplementary Fig. 13**.

Supplementary Fig. 13. Rheological properties of Sylgard186, Dragonskin 30, and epoxy. **a** Viscosity under different shear rates. **b** Storage and loss moduli under different shear stresses.

To alleviate your concerns, on Page 11, we add:

“Note that only representative inks are shown in Fig. 4a, b while rheological measurements of other inks are given in Supplementary Fig. 13.”

Also, Sylgard 184, 186, Dragonskin 30, 10, Ecoflex 00-30, 00-10 are widely used commercial Silicone-based polymers. The composition of Part A and B of Sylgard 184 is provided in Supplementary Fig. 2 (See **Response 6**), while composition of other inks is not available/or disclosed due to the commercial proprietary reasons. To alleviate your concerns, we provide as more details as we can in the Materials and Methods.

On Page 14, we add:

Pure thermosetting ink. Sylgard 184 and Sylgard 186 are purchased from Dow Corning, USA. Dragonskin 30, Dragonskin 10, Ecoflex 00-30, and Ecoflex 00-10 are purchased from Smooth-on, USA. The silicone ink was prepared by directly blending corresponding Part A and B at a specific mass ratio in a 50 ml beaker. For Sylgard 184 and 186, Part A: B=10:1. For Dragonskin 30, Dragonskin 10, Ecoflex 00-30, and Ecoflex 00-10, Part A: B=1:1. For Dragonskin 10 + 20 wt%SiO₂ and blended using a planetary mixer (ARV-310, Thinky,

Japan) at 2000 rpm for 5 min. The epoxy ink is purchased from Sinoepc China, and mixing E39D and E20 with 1:1 weight ratio by manual stirring under heating with a heat gun, then added 14.2 phmr 2-ethyl-4-methylimidazole (Aladdin, China) as curing agent.

Comment #8

The low resolution of Fig 2 & Fig 3 makes some key information in these Figures confusing.

Response #8

We are sorry for the confusion. We have updated Fig. 2 and Fig. 3 and uploaded high-resolution images. We have provided revised Fig. 2 in our response 3. Revised Fig. 3 is attached here for your convenience.

Fig. 3. Characterization of ISDH-enabled DIW of low-viscosity Sylgard 184 with base-to-curing agent ratio of 10:1. **a** Photograph of printing a cylinder tube. **b** Optical and scanning electron microscope (SEM) images of a printed cylinder tube with a height of 120 mm and thickness of 1.5 mm. (Left scale bar=12 mm, right scale bar=0.8 mm). **c** Optical image of a printed wrinkled ring with a diameter of 90 mm using a nozzle with 1-mm diameter (scale bar=2 mm). **d** Serpentine network printed with a 25 μm nozzle (scale bar=100 μm). **e-g** Comparison of the (e) tensile test; (f) elastic modulus and ultimate strength; (g) infrared and (h) Raman spectroscopy between ISDH-printed and molded samples.

Comment #9

Equation 1 has different names throughout this article, including “(1)” (near where the equation itself), “Eq. 1”, “Equation 1” and “equation (1)”.

Response #9

We thank you for pointing this out. We have thoroughly gone through the paper and unified formats as Eq. 1.

In summary, we hope these revisions have addressed the concerns of three reviewers and significantly improved the quality of our work, making it suitable for publication in *Nature Communications*.

Reviewers' Comments:

Reviewer #1:

Remarks to the Author:

The reviewers have done an excellent and thorough job in responding to the critiques and suggestions and strengthened the manuscript considerably.

Reviewer #2:

Remarks to the Author:

The authors have addressed my comments and the paper can be accepted.

Reviewer #3:

Remarks to the Author:

This reviewer thanks the efforts from the authors in revising the manuscript and answering questions from the reviewers. First of all, this reviewer acknowledges that this manuscript has been greatly revised. The abstract and introduction were almost completely rewritten (>80%?) with the inappropriate statements revised. In addition to that, the newly added demonstrations using this strategy further indicate the wide applicability of this method in manufacturing functional materials, such as magnetic stent and magnetic powered LED, which would have potential applications in biomedical engineering and soft robotics. Hence, the reviewer feels that the authors have appropriately answered the questions and the current version of manuscript is of high scholarship, given the fact that this concept of this work is novel.

However, this reviewer is concerned that the contents the authors summarized in Supplementary Table 2 did not cover representative categories in DIW of thermosets. The authors mentioned four types of DIW besides this work (ISDH). However, there are still a lot of other works that do not fall into any of (1) Rheology modification by adding particles; (2) Embedded printing in bath support; (3) Frontal polymerization by chemical reaction. These works utilize well-designed molecular entities and the non-covalent interactions between them to achieve DIW 3D printing. Just to name a few, self-assembled bottlebrush copolymers are DIW 3D printable and UV curable even without particles added (Science Advances 2020, 6, eabc6900.). Some liquid crystalline polymers do not require additional particles to be DIW 3D printed (Advanced Materials 2018, 30, 1706164). It would be highly appreciated if the authors could have those representative works highlighted in the table to better represent the state-of-the-art.

Responses to Comments on “NCOMMS-22-31470A”

Again, we sincerely thank three reviewers for reading our revised manuscript. Your constructive comments and suggestions are fully addressed point-by-point in this letter. For your convenience, the corresponding modifications are marked in blue in the revised manuscript.

Reviewer # 1

General comments.

The reviewers have done an excellent and thorough job in responding to the critiques and suggestions and strengthened the manuscript considerably.

Our response

We thank the reviewer for accepting the manuscript for publication.

Reviewer # 2

General comments.

The authors have addressed my comments and the paper can be accepted.

Our response

We thank the reviewer for accepting the manuscript for publication.

Reviewer #3

General Comments.

This reviewer thanks the efforts from the authors in revising the manuscript and answering questions from the reviewers. First of all, this reviewer acknowledges that this manuscript has been greatly revised. The abstract and introduction were almost completely rewritten (>80%?) with the inappropriate statements revised. In addition to that, the newly added demonstrations using this strategy further indicate the wide applicability of this method in manufacturing functional materials, such as magnetic stent and magnetic powered LED, which would have potential applications in biomedical engineering and soft robotics. Hence, the reviewer feels that the authors have appropriately answered the questions and the current version of manuscript is of high scholarship, given the fact that this concept of this work is novel.

Our response

We thank the reviewer for the favorable comment.

Comment #1

However, this reviewer is concerned that the contents the authors summarized in Supplementary Table 2 did not cover representative categories in DIW of thermosets. The authors mentioned four types of DIW besides this work (ISDH). However, there are still a lot of other works that do not fall into any of (1) Rheology modification by adding particles; (2) Embedded printing in bath support; (3) Frontal polymerization by chemical reaction. These works utilize well-designed molecular entities and the non-covalent interactions between them to achieve DIW 3D printing. Just to name a few, self-assembled bottlebrush copolymers are DIW 3D printable and UV curable

even without particles added (Science Advances 2020, 6, eabc6900.). Some liquid crystalline polymers do not require additional particles to be DIW 3D printed (Advanced Materials 2018, 30, 1706164). It would be highly appreciated if the authors could have those representative works highlighted in the table to better represent the state-of-the-art.

Our response

We thank the reviewer for the suggestions. We have added the suggested two references (ref # 30 and 41) to **Supplementary Table 2** (see Next page) and categorized them into self-assembled yield-stress polymers and in situ light-curing. In addition, we added another two references [(1) Three-Dimensional Printable, Extremely Soft, Stretchable, and Reversible Elastomers from Molecular Architecture-Directed Assembly. Chemistry of Materials 33, 2436-2445 (2021); (2) Three-dimensional printing of functionally graded liquid crystal elastomer. Science advances 6, eabc0034 (2020)] to strengthen your point. The following discussions are added to the revised Introduction.

On Page 3, we add:

“Other approaches for 3D printing thermosets are based on the direct ink writing (DIW) technique¹⁹⁻²². Usually, successful DIW requires a yield-stress ink to form self-support. For example, some self-assembled bottlebrush polymers^{23,24} can be directly printed. But for many low-viscosity thermosets, DIW is often difficult due to structural collapse.”

On Page 4, we add:

“Last but not least, some photo-curable thermosetting inks (even with low viscosity) may also be directly written by utilizing *in situ* light-curing during the DIW process. For instance, some liquid crystalline polymers can be printed under UV light^{41,42}. Similar to DLP and SLA, however, this 3D printing strategy may only apply to light-curable thermosets.”

Supplementary Table 2. Comparison of different approaches for 3D printing of thermosets

Type	Mechanism	Minimum nozzle /resolution (µm)	Printable height	Material applicability	Tailored thermosets property	Multi-functionality	Hybrid printing
Vat polymerization	Direct sound printing by sonochemical reaction ¹⁸	None/280	High	Limited inks with specific porosity	No	No	No
	DLP or SLA by light-curing ¹⁴⁻¹⁷	None/20	High	Light-curable resin	No	Yes	Yes
Direct ink writing	Rheology modification by adding particles ²³⁻²⁹	50/50	Low	Yield-stress	Yes	Yes	Limited
	Self-assembled yield-stress polymers ^{23,24}	150/150	Low	Yield-stress	Yes	No	Limited
	Embedded printing in bath-support ³³⁻³⁶	100/100	High	Newtonian, shearing-thinning, yield-stress	No	Limited	No
	Frontal polymerization by chemical reaction ³⁸⁻⁴⁰	250/250	High	Limited to dicyclopentadiene	Yes	No	Limited
	In situ light-curing ^{41,42}	100/100	High	Light-curable resin	No	Yes	Yes
	In situ gelation by ISDH (This work)	25/50	High	Newtonian, shearing-thinning, yield-stress	No	Yes	Yes